# PRDM16 determines specification of ventricular cardiomyocytes by suppressing alternative cell fates

Jore Van Wauwe[1], Alexia Mahy[1], Sander Craps[1], Samaneh Ekhteraei-Tousi[2], Pieter Vrancaert[1],
Hannelore Kemps[1], Wouter Dheedene[1], Rosa Doñate Puertas[2], Sander Trenson[3], H. Llewelyn Roderick[2],
Manu Beerens[4,5], Aernout Luttun[1]

PRDM16 is a transcription factor with histone methyltransferase activity expressed at the earliest stages of cardiac development. Pathogenic mutations in humans lead to cardiomyopathy, conduction abnormalities, and heart failure. PRDM16 is specifically expressed in ventricular but not atrial cardiomyocytes, and its expression declines postnatally. Because in other tissues PRDM16 is best known for its role in binary cell fate decisions, we hypothesized a similar decision-making function in cardiomyocytes. Here, we demonstrated that cardiomyocyte-specific deletion of *Prdm16* during cardiac development results in contractile dysfunction and abnormal electrophysiology of the postnatal heart, resulting in premature death. By combined RNA+ATAC single-cell sequencing, we found that PRDM16 favors ventricular working cardiomyocyte identity, by opposing the activity of master regulators of ventricular conduction and atrial fate. Myocardial loss of PRDM16 during development resulted in hyperplasia of the (distal) ventricular conduction system. Hence, PRDM16 plays an indispensable role during cardiac development by driving ventricular working cardiomyocyte identity.

## Introduction

The mammalian heart is a complex organ, composed of four morphologically, molecularly, and functionally diverse chambers. Atria and ventricles each host cardiomyocytes (CMs) with distinct expression profiles and different electrophysiological properties (Cao et al, 2023; Kane & Terracciano, 2017; Ng et al, 2010). The atria eject blood into the ventricles, while the latter pump the blood through our lungs and body. This pumping activity is ensured by ventricular working CMs, which continuously contract in sequence from the cardiac apex to its base upon electrical stimulation. This pattern is coordinated by cells from the cardiac conduction system, which display different features than the contractile working CMs (Munshi, 2012). The ventricular conduction system (VCS) is responsible for the rapid propagation of the electrical signal through the ventricles. The proximal VCS encompasses the bundle of His located in the septum, which splits at the base of the heart in the left and right bundle branches. These bundle branches each ramify in a complex network of subendocardial Purkinje fibers (PFs), known as the distal VCS (Choquet et al, 2021). Although differentiation of the proximal VCS is complete at ventricular septation, the formation of the distal Purkinje system occurs in two phases whereby an initial scaffold is formed early on, followed by a phase of recruitment of additional PFs by continuous differentiation from trabecular precursors, a process that persists until birth (Fig S1A; all supplemental online items are designated "S") (Mikawa et al, 2003; Choquet et al, 2021). The trabecular protrusions gradually disappear, whereas the fast proliferating compact myocardial layer expands and both the ventricular wall and VCS mature (MacGrogan et al, 2018; Lupu et al, 2020). Defects in the specification and morphogenesis of the VCS cause life-threatening syndromes, including Brugada and long-QT syndromes, both characterized by lethal cardiac arrhythmias (Haissaguerre et al, 2016; Choquet et al, 2021). Hence, it is important to understand the cellular and molecular mechanisms that underlie the specification and formation of cardiac wall development.

This heterogeneity among CMs is largely established by cell fate decisions governed and monitored by transcription factor (TF) networks that tightly orchestrate spatial and temporal regulation of gene expression, through either DNA interactions or chromatin remodeling (Christoffels & Moorman, 2009; Shekhar et al, 2018; Pawlak et al, 2019). Over the last decades, many TFs involved in such decisions in CMs have been identified, including TBX5, a master regulator of atrial and conduction cell fates (Pawlak et al, 2019; Choquet et al, 2021; Cao et al, 2023). Positive regulatory domain–containing protein (PRDM)16, a member of the PRDM family of TFs

---

[1]Center for Molecular and Vascular Biology, Department of Cardiovascular Sciences, KU Leuven, Leuven, Belgium [2]Laboratory of Experimental Cardiology, Department of Cardiovascular Sciences, KU Leuven, Leuven, Belgium [3]Cardiology Lab, Department of Cardiovascular Sciences, KU Leuven, Leuven, Belgium [4]Institute for Clinical Chemistry and Laboratory Medicine, Medizinische Klinik und Poliklinik Universitätsklinikum Hamburg-Eppendorf, Hamburg, Germany [5]German Centre of Cardiovascular Research (DZHK), Partner Site Hamburg, Luebeck, Kiel, Hamburg, Germany

Correspondence: aernout.luttun@kuleuven.be

with methyltransferase activity, is a TF specifically expressed in ventricular CMs while absent in atria (Bjork et al, 2010; Arndt et al, 2013; Wu et al, 2022). Its asymmetric expression pattern in the heart suggests a cell fate decision-making role. Indeed, PRDM16 was recently put forward as a positive regulator of ventricular compact CM fate, highlighting for the first time the contribution of PRDM16 to the specification and heterogeneity among CMs (Wu et al, 2022). Accordingly, pathogenic mutations lead to left ventricular non-compaction cardiomyopathy (LVNC), dilated cardiomyopathy, and ventricular conduction abnormalities in patients (Arndt et al, 2013; Harms et al, 2014; Long et al, 2017; van Waning et al, 2018; Mazzarotto et al, 2021; Wu et al, 2022; Kramer et al, 2023). Loss-of-function studies in mice resulted in LVNC, ventricular conduction defects, and hypertrophic cardiomyopathy (Cibi et al, 2020; Nam et al, 2020; Kang et al, 2022; Wu et al, 2022; Kramer et al, 2023; Sun et al, 2023; Kuhnisch et al, 2024). However, as a result of using different Cre driver strains to induce PRDM16 loss, the phenotypic outcome of these animal studies was highly heterogeneous ranging from early postnatal lethality to cardiac dysfunction only acquired in adulthood or after an additional challenge (Cibi et al, 2020; Nam et al, 2020; Wu et al, 2022; Kramer et al, 2023). In these studies, expression profiling has been performed either before birth or during the adult stage. Therefore, the transcriptional changes governed by PRDM16 during early postnatal cardiac development, when myocardium is still maturing and cell fate decisions need to be maintained, remain poorly characterized (Li et al, 2022; Sweat et al, 2023).

We and others have previously shown the asymmetric expression pattern of PRDM16 in endothelial cells (ECs) and adipocytes, which coincides with its role in cell fate decision-making. PRDM16 favors an arterial EC and brown adipocyte cell fate over venous ECs and white adipocytes, respectively (Seale et al, 2007; Aranguren et al, 2013; Craps et al, 2021; Thompson et al, 2023; Van Wauwe et al, 2024 Preprint). Mechanistic studies further revealed that PRDM16 not only exerts its effects on gene expression, and hence cell fate decisions, via direct binding to the promoter regions of its target genes, but primarily indirectly via interaction with other DNA binding TFs and by altering the accessibility of chromatin through its methylating activity or recruitment of chromatin-modifying enzymes (Kajimura et al, 2008; Harms et al, 2014; Li et al, 2015; Cibi et al, 2020; Jiang et al, 2022; Wu et al, 2022). It, however, remains unknown whether and how PRDM16 orchestrates cardiac development and function through modification of chromatin accessibility, rather than transcriptional activity, at defined loci.

Here, we applied a combined RNA and Assay for Transposase-Accessible Chromatin (ATAC) sequencing approach at single-cell resolution to resolve cellular and molecular mechanisms governed by PRDM16 in heart development at 7 d after birth. We found that in accordance with its asymmetric myocardial expression pattern, PRDM16 is involved in a decision process whereby it favors ventricular contractile CM cell fates through opposing the activity of master regulators of atrial and conduction cell fates. Myocardial deletion of Prdm16 during development resulted in hyperplasia of the (distal) VCS, providing a potential explanation for the sudden death of PRDM16-deficient mice we observed between 1 and 3 wk after birth.

# Results

## PRDM16 loss in CMs during development causes early-onset cardiomyopathy

PRDM16 expression in the developing mouse heart was evident in the compact and trabecular ventricular myocardium from embryonic day (E)10.5 onward, but was notably absent from the atria, as previously reported (Fig S1A–D) (Bjork et al, 2010; Arndt et al, 2013; Litvinukova et al, 2020; Wu et al, 2022). At E14.5 and E17.5, PRDM16 expression became more restricted to CMs of the compact myocardium (Fig S1E and F). Immunostaining at postnatal day (P)7 revealed that PRDM16 expression in the ventricles was present in CMs and in coronary arterial ECs and smooth muscle cells (SMCs) (Fig S1G and H). Notably, the expression of PRDM16 in the heart significantly declined after birth, as shown by Wu et al (2022). To study the role of PRDM16 during cardiac development, we generated mice with cardiomyocyte-specific Prdm16 deletion from early cardiac development onward, by inter-crossing an Sm22α-Cre driver line with mice harboring two floxed Prdm16 exon 9 alleles (Fig S2A), resulting in $Prdm16^{lox/lox};Sm22α-Cre^{Tg/+}$ or $Prdm16^{lox/lox};Sm22α-Cre^{+/+}$ offspring (referred to as $Prdm16^{cKO}$ mice and $Prdm16^{WT}$ littermate controls, respectively). Sm22α is transiently expressed in the mouse (pre)myocardium between E8.0 and E12.5 (Fig S1A) (Li et al, 1996). Sm22α-Cre driver activity faithfully reported the endogenous Sm22α expression pattern in eGFP reporter mice, as evident from eGFP expression throughout the myocardium (Fig S2B and C). At P7, immunohistochemistry, RT–qPCR, and immunoblotting demonstrated that our strategy successfully eliminated PRDM16 expression in CMs (with an efficiency at P7 of 93% ± 1%, n = 4; Fig S1I, K–M). PRDM16 loss was also evident in coronary artery SMCs and arterial SMCs in P7 brains and lungs, but not in arterial ECs, bronchiolar epithelial cells, and various cells in the choroid plexus known to express PRDM16 (Figs S1J and S2D–F) (Shimada et al, 2017; Strassman et al, 2017; Fei et al, 2019; Craps et al, 2021).

In contrast to the 100% mortality by P7 previously reported for Prdm16 deletion in CMs from E7.5 onward by cTnT-Cre and Xmlc2-Cre (Wu et al, 2022), we did not observe notable losses of $Prdm16^{cKO}$ pups at that time, as shown by a nearly Mendelian distribution (Table 1). However, $Prdm16^{cKO}$ mice had lower body weights (Table S1), a significantly reduced ejection fraction (EF), and CM hypertrophy compared with their $Prdm16^{WT}$ littermates (Fig 1A and B and Table S1). $Prdm16^{cKO}$ LVs featured significant up-regulation of stress and hypertrophy-related marker genes, that is, Nppa and Nppb, encoding the atrial and brain natriuretic peptide, respectively (Fig 1C). Unlike their $Prdm16^{WT}$ littermates, $Prdm16^{cKO}$ offspring had clear signs of perivascular and interstitial fibrosis (Fig 1D). Like in humans and zebrafish (Arndt et al, 2013; Hong et al, 2014; van Waning et al, 2018; Kramer et al, 2023), $Prdm16^{cKO}$ pups showed significant signs of a reduced compact LV, together with excessive trabeculation (Figs 1E and S3). Moreover, already at P7, $Prdm16^{cKO}$ pups showed an aberrant electrocardiogram (ECG), that is, a prolonged QRS duration (Fig 1F), as previously described at later time points in mice and humans (Hong et al, 2014; Nam et al, 2020), and a

**Table 1. Genotype distribution.**

| Genotype | Prdm16^WT | Prdm16^cKO |
|---|---|---|
| Seven days of age (P7) | | |
| Expected ratio (%) | 50 | 50 |
| Observed ratio (%) | 52 | 48 |
| Observed absolute | n = 42 | n = 39 |
| Post-weaning (all) | | |
| Expected ratio (%) | 50 | 50 |
| Observed ratio (%) | 71 | 29 |
| Observed absolute | n = 622 | n = 253 |
| Post-weaning (male) | | |
| Expected ratio (%) | 50 | 50 |
| Observed ratio (%) | 71 | 29 |
| Observed absolute | n = 334 | n = 135 |
| Post-weaning (female) | | |
| Expected ratio (%) | 50 | 50 |
| Observed ratio (%) | 71 | 29 |
| Observed absolute | n = 288 | n = 118 |

significantly increased QRS amplitude that accords with LV hypertrophy. Although our Sm22α-Cre driver also eliminated PRDM16 expression in SMCs (Fig S1H versus Fig S1J), SMC coverage was not affected in *Prdm16^cKO* hearts (Fig S1N), suggesting that the earlier PRDM16 loss in CMs (Fig S1A) was the primary culprit for the observed phenotypic aberrations. Thus, deficiency of PRDM16 in CMs caused severe signs of cardiomyopathy early on.

### *Prdm16* deletion in CMs during development leads to premature death or progressive cardiomyopathy

Although nearly all *Prdm16^cKO* mice were alive at P7, 60% of both male and female *Prdm16^cKO* mice died by weaning age (3 wk), after which no further losses were observed (Table 1). We then monitored the cardiac phenotype of the surviving adult mice, which revealed progressive signs of heart failure to a similar extent in both male and female mice (Fig S4 and Table S1). Both at 8 and at 16 wk of age, *Prdm16^cKO* mice showed significant diastolic and systolic dysfunction, as evidenced by an increased E/e′ ratio and a lowered EF, respectively (Fig S4A and E and Table S1). *Prdm16^cKO* hearts had dilated LVs, as evidenced from significantly increased LV internal diameter and decreased posterior wall thickness obtained at end-systole (LVIDs and LVPWs, respectively; Table S1), which was more pronounced at 16 wk of age. *Nppa* and *Nppb* expression was significantly up-regulated in the LVs of adult *Prdm16^cKO* mice (Fig S4B and F). Furthermore, *Prdm16^cKO* mice also featured significant perivascular fibrosis at 8 and 16 wk, where the latter time point also featured significant interstitial fibrosis (Fig S4C and G). Like P7 pups, surviving adult *Prdm16^cKO* mice showed a significantly prolonged QRS duration and an increased QRS amplitude, while maintaining a normal heart rate (Fig S4D and H and Table S1).

### PRDM16 loss in CMs during development perturbs the cardiac cellular landscape

Because *Prdm16^cKO* mice started to die beyond P7 and this time point has recently been proposed to represent a transition state that is essential for CM subtype specification and subsequent maturation (Li et al, 2022), we chose this transition state to look into the effect of PRDM16 on the cellular composition of the heart by single-nucleus (sn)RNA/ATACseq (Fig 2A). We chose to focus our analysis on the LV only given the restricted expression of PRDM16 in the ventricles (our study and those by others [Bjork et al, 2010; Arndt et al, 2013; Wu et al, 2022]) and because of the predominant effect of pathogenic *PRDM16* mutations leading to LVNC in patients (Arndt et al, 2013; Delplancq et al, 2020; Mazzarotto et al, 2021). After pooling four samples per condition and stringent quality control (Table S2), 5,468 nuclei remained. Unsupervised low-resolution clustering and dimensionality reduction on the integrated *Prdm16^cKO* and *Prdm16^WT* RNA/ATACseq datasets resolved the major cell types of the LV, that is, 2 clusters of CMs including *Myh6^+* mature (m)CMs and *Top2a^+* proliferating (p)CMs, and additional non-CM cell types including *Dcn^+* fibroblasts (FBs), *Cdh5^+* ECs, *Pdgfrb^+* mural cells (MCs), and *Csf1r^+* immune cells (ICs; Fig 2A and B) (Wei et al, 2011; Hu et al, 2018; Asp et al, 2019; Cui et al, 2020; Muhl et al, 2020; Tucker et al, 2020). Although the proportions of these fractions closely reflected the expected size at this postnatal stage in *Prdm16^WT* hearts (Hu et al, 2018; Cui et al, 2020), the relative contributions of these populations shifted in *Prdm16^cKO* hearts (Fig 2C). Indeed, in accordance with the observed fibrosis, the proportion of FBs increased significantly and their expression profile changed toward a myofibroblast-like signature (Table S3), whereas mCM nuclei were significantly reduced in numbers, as confirmed by in situ assessment by immunostaining on an independent set of mice (Fig 2D). Smaller shifts were seen in the vascular cellular landscape, with a slight increase in the EC compartment and a decrease in the MC compartment (Fig 2C).

To gain a greater insight into the cellular landscape shifts, including the identification of different CM subpopulations featuring unique marker genes, we re-clustered the total CM nucleus population (Fig S5A and Table S3). We now found 3 CM subclusters, of which the largest subcluster (0) expressed several markers of mature ventricular CMs (including *Ryr2*; Fig S5A and Table S3) (Pallante et al, 2010; Goodyer et al, 2019; Cao et al, 2023). Interestingly, 90.3% of *Prdm16^WT* nuclei were represented in this larger mature CM subcluster. CM subcluster (1) represented a "mixed" nature as it was marked not only by genes known to be typical for the VCS (e.g., *Slc6a6*, *Cacna2d2*, and *Ryr3*; Table S3) but also by genes known as atrial CM genes (e.g., *Myl4* and *EphA4*; Table S3) (Goodyer et al, 2019; Cao et al, 2023). *Prdm16^WT* nuclei only represented 1.2% of this atrial/VCS cluster. Finally, the smallest CM subcluster (2) exhibited a very clear "proliferation" gene signature, including *Top2a*. We also re-clustered the other main cell types (Fig S5A and Table S3) and examined the expression of *Prdm16* within each subcluster. This analysis detected the presence of *Prdm16* in *Prdm16^WT* nuclei in all CM subclusters, as well as in the SMCs and in arterial ECs, in accordance with the expression pattern we documented by IF staining above (Figs S1G and H, S2D-F, and S5B).

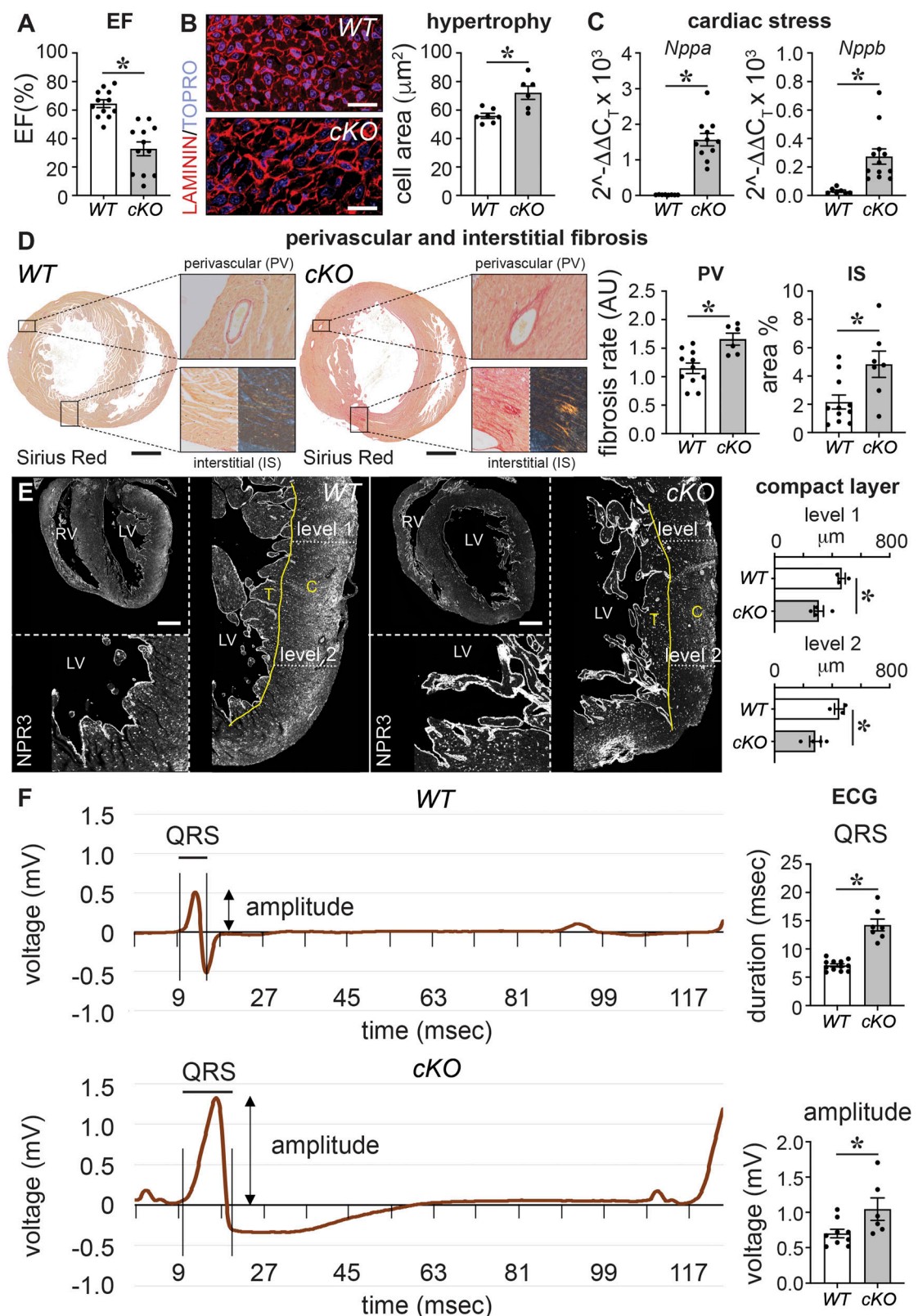

**Figure 1. PRDM16 loss during cardiac development causes early-onset cardiomyopathy.**
**(A)** Ejection fraction (EF) of 7-d-old (P7) mouse pups expressed in % (n = 12/12). WT, wild-type; cKO, Prdm16 conditional knockout. **(B)** Representative pictures of cross-sections stained with LAMININ (red) and TO-PRO-3 (blue) and quantitative analysis of the cardiomyocyte size of P7 hearts expressed in μm² (n = 7/6). **(C)** mRNA levels of cardiac stress markers Nppa (n = 8/11) and Nppb (n = 7/12), measured in P7 heart apex. **(D)** Representative images of Sirius Red–stained cross-sections revealing fibrosis in

Interestingly, the overall loss of CMs in *Prdm16^cKO* mice could mainly be attributed to a remarkable 2.7-fold reduction of the mature CM *Prdm16^cKO* cluster, whereas the subcluster with a mixed atrial/VCS signature was strikingly overrepresented in *Prdm16^cKO* hearts (i.e., ~30-fold increase; Fig S5A). The latter may partly account for the aberrant ECG of *Prdm16^cKO* mice (Fig 1F). Altogether, deletion of *Prdm16* during ventricular wall development altered the cellular landscape of the heart aligning with the aberrant cardiac phenotype.

### PRDM16 loss in CMs triggers changes in gene expression and chromatin accessibility related to hypertrophy, metabolism, conduction, and TGFβ signaling

We next performed a differential analysis of RNA expression and chromatin accessibility between *Prdm16^cKO* and *Prdm16^WT* CMs to better understand the molecular mechanisms driving the observed cardiac phenotypes. It was remarkable that ~79% of the mature CM cluster 0 represented *Prdm16^WT* nuclei, whereas ~97% of nuclei from the "mixed" cluster originated from *Prdm16^cKO* hearts (Fig 3A and B). In contrast, *Prdm16^cKO* did not affect the size of the cluster representing proliferating CMs (Fig 3B). Because the main CM subclusters were almost exclusively related to one of the genotypes (and hence the contamination of cells from the other genotype would have a limited impact on differential gene expression), we decided to look at differentially expressed genes (DEGs) and differentially accessible regions (DARs) of the CM cluster in total, that is, by comparing *Prdm16^cKO* versus *Prdm16^WT* CMs (Fig 3B). This revealed 1,665 DEGs (1,137 and 528, higher or lower expressed in *Prdm16^cKO* mice, respectively; Fig 3C and Table S4). Upon screening of the DEG lists, a number of clear traits caused by PRDM16 deficiency were manifest. First, in accordance with the observed CM size increase (Fig 1B), higher expressed DEGs were associated with cardiac hypertrophy (e.g., *Nppa*, *Gpx3*, *Sparc*; Fig 3C and Table S4) (Farrell et al, 2018; Cibi et al, 2020; Vigil-Garcia et al, 2021). Second, lower expressed genes were related to fatty acid (FA) metabolism (e.g., *Lpl*, *Ppara*, *Cpt1a*; Fig 3C and Table S4), reflecting an energy source switch in these cells of the failing heart away from FA. Third, the top higher (i.e., *Fgf12*, encoding a non-canonical FGF that does not bind FGF receptors but associates with sodium channels) and lower expressed (i.e., *Kcnd2*, encoding the potassium channel subunit $K_v4.2$) genes are both related to ion channels, potentially involved in the aberrant cardiac conduction (Table S4). Finally, in line with previous findings (Arndt et al, 2013; Kodo et al, 2016; Nam et al, 2020; Sun et al, 2023), we also observed higher expression in (target) genes from the increased TGFβ signaling pathway (e.g., *Spred1*, *Smad2*, *Tgfb2*; Fig 3C and Table S4); however, a similar proportion of (target) genes was lower expressed (e.g., *Ltbp1*, *Itgb6*, *Tgfbr3*; Fig 3C and Table S4).

Likewise, we identified 718 DARs, of which 501 and 217 were more and less accessible in *Prdm16^cKO* mice, respectively (Fig 3D and Table S5). Annotation of altered chromatin regions revealed that this occurred mostly at intronic and intergenic, but also at promoter locations (Fig 3D). As more DEGs were higher expressed and more DARs displayed an open chromatin structure in the absence of PRDM16, our results suggest that PRDM16 mainly acts as a transcriptional repressor in CMs. Gene ontology (GO) analysis on the more highly expressed DEGs and on the open DARs revealed enrichment for GO terms related to "heart growth" and "(dilated) cardiomyopathy," and interestingly also identified terms associated with "chamber type," "conduction system," and "arrhythmias," the last two compatible with the aberrant ECG (Fig 3E and Table S6). On the contrary, many GO terms related to fat(ty acid) metabolism/oxidation were identified among the down-regulated DEGs and genes associated with more closed DARs (Table S6). Additional annotation analysis using ToppGene confirmed these findings including the appearance of "cardiac conduction system development" and "bundle of His to Purkinje myocyte signaling" among the functional terms associated with higher expressed DEGs, and the extraction of "fatty acid oxidation" and "heart development" and "regulation of cardiac muscle contraction" as functional terms related to the lower expressed genes (Table S6). Altogether, PRDM16 loss caused expression/chromatin accessibility changes in CMs in line with our observed cardiac phenotype and previous studies.

### PRDM16 loss in CMs causes a shift toward atrial and conduction cell fates

A deeper analysis of the DEGs unveiled that many of the up-regulated DEGs encoded known atrial markers (e.g., *Fgf12*, *Myl4*; Fig 4A and Table S4) or (ventricular) conduction markers (i.e., *Ryr3*, *Cacna2d2*; Fig 4B and Table S4) (Hartung et al, 1997; Koibuchi & Chin, 2007; Ng et al, 2010; Wiencierz et al, 2015; DeLaughter et al, 2016; Kane & Terracciano, 2017; Litvinukova et al, 2020; Nam et al, 2020; Tucker et al, 2020; Wu et al, 2022). In support of this fate switch, we compared our DEGs with published gene expression signatures of CMs from atria (Cao et al, 2023) or the conduction system (Shekhar et al, 2018). We found that 312 and 287 out of 1,665 DEGs were part of the published atrial or conduction system signatures, of which the majority (273 or 87.5% and 246 or 85.7%, respectively) were higher expressed after *Prdm16* deletion (Fig 4C and D and Tables S4 and S7). Because the published atrial and conduction signatures showed some overlap, we filtered out these common genes after which 216 and 191 unique atrial and unique conduction DEGs remained. Again, most of the unique marker genes (i.e., 84.7% of atrial and 81.7% of conduction DEGs) were significantly higher expressed in *Prdm16^cKO* CMs, demonstrating that there was a

P7 mouse hearts, insets showing perivascular (PV; brightfield, *top*) and interstitial (IS; brightfield, *bottom left*; polarized light, *bottom right*) fibrosis, and bar graphs showing the quantitative analysis of PV and IS fibrosis. PV fibrotic area was corrected for the smooth muscle cell area of the vessel and expressed in arbitrary units (AU; *n* = 11/7). **(E)** Representative images of Natriuretic Peptide Receptor 3–stained transversal sections marking the endocardial lining and quantitative analysis of the compact myocardial wall thickness expressed in μm (*right*); level 1 represents the base, and level 2 represents the apex of the left ventricle (LV; *n* = 3/4). The yellow line delineates the trabecular (T)/compact (C) border. RV: right ventricle. **(F)** Average surface electrocardiogram measured in P7 mice in rest expressed in mV over time (in msec). Bar graphs show quantitative analysis of QRS duration (*n* = 12/7) and amplitude (*n* = 9/6). Quantitative data are expressed as the mean ± SEM; *P < 0.05 by a *t* test. Scale bars: 20 μm (B), 500 μm (D, E).

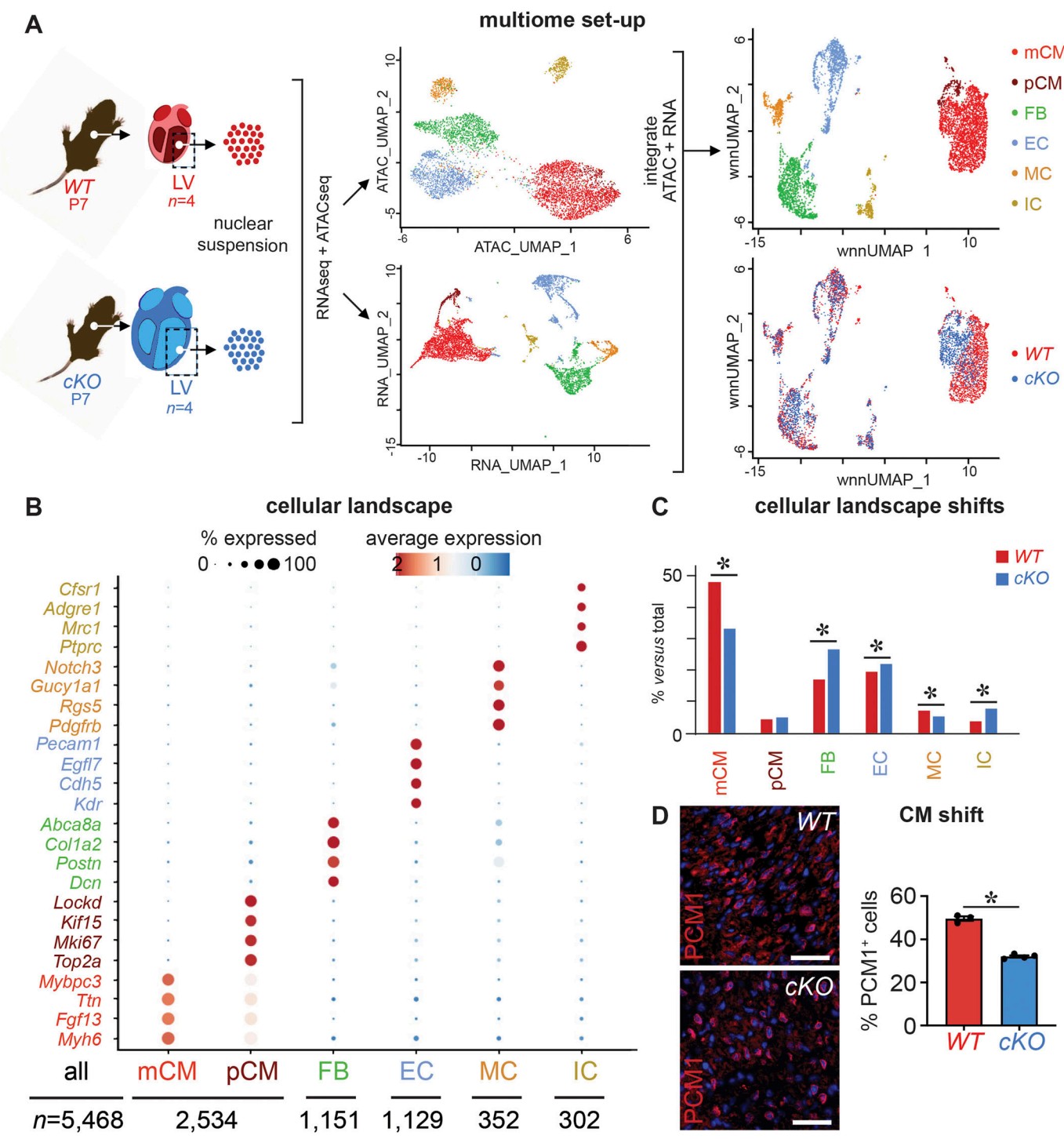

**Figure 2. PRDM16 loss in CMs during development perturbs the cardiac cellular landscape.**
**(A)** Experimental setup of droplet-based single-nucleus multiome RNA and ATAC sequencing (10x Genomics) experiment on pooled left ventricles (LVs) of 7-d-old (P7) *WT* (*n* = 4) or *Prdm16* conditional knockout (*cKO*; n = 4) mouse hearts. Nuclei were isolated and subjected to combined single-nucleus RNA and ATAC sequencing (*left*). Uniform Manifold Approximation and Projection dimensional reduction panels are shown for each separate modality (*middle*), as well as after integration (*top right*) and splitting per genotype (*bottom right*). **(B)** Dot plot representing major cell populations (clusters) identified in the heart on the x-axis with their marker genes represented on the y-axis. Color code of different clusters matches that of the integrated Uniform Manifold Approximation and Projection in (A). The size of the dots represents the percentage of cells expressing the marker gene; the dot color indicates average expression levels expressed in log(fold change). The number of nuclei per cluster is indicated below the plot. **(C)** Bar graph representing the cellular proportions for each cluster in *WT* versus *cKO* samples. mCM, mature cardiomyocyte; pCM, proliferating cardiomyocyte; FB, fibroblast; EC, endothelial cell; MC, mural cell; IC, immune cell. To calculate marker genes, a Wilcoxon test was used with log(fold change) threshold = 0, $P_{adjusted}$ < 0.05; to calculate cell proportion differences, Fisher's exact test was used with FDR < 0.05. **(D)** Representative images of PCM1-stained cross-sections of *WT* and *cKO* P7 hearts (*left*) and corresponding quantification of cell proportion expressed as % (*right*; *n* = 3/4). Quantitative data are expressed as the mean ± SEM; *P < 0.05 by a *t* test. Scale bars: 20 *µm* (D).

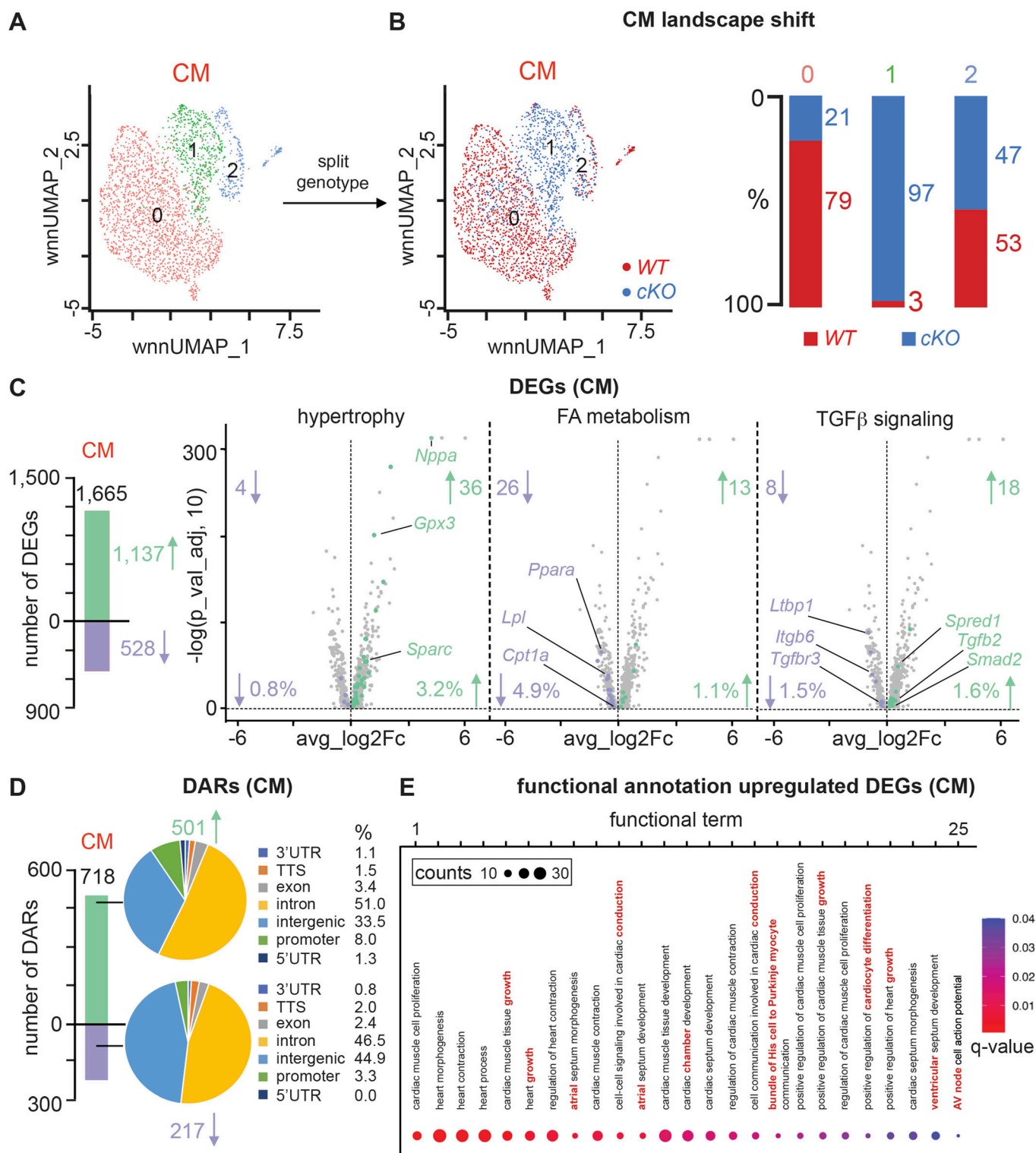

**Figure 3. PRDM16 loss in CMs triggers changes in gene expression and chromatin accessibility related to hypertrophy, metabolism, conduction, and TGFβ signaling.**
**(A, B)** Uniform Manifold Approximation and Projection (UMAP) representing re-clustered cardiomyocyte (CM) cellular landscape identifying three major cell clusters (A) and the same UMAP split by genotype ((B), *left*; red: *WT* CMs; blue: *Prdm16*-deficient CMs [*cKO*]). **(A, B)** Bar graph ((B), *right*) shows proportions of *WT* (red) and *cKO* (blue) CMs represented in subclusters 0, 1, and 2 from panel (A). **(C)** Bar graph showing the numbers of differentially expressed genes (DEGs) identified in CMs. Green and purple bars represent the number of higher and lower expressed genes, respectively. Volcano plots show higher expressed (green) or lower expressed (purple) DEGs in CMs,

double fate shift upon *Prdm16* deletion (Fig S6A and B and Tables S4 and S7). Vice versa, CMs from *Prdm16^cKO* hearts displayed significantly reduced expression levels of genes characteristic of ventricular (Cao et al, 2023) or working CMs (Shekhar et al, 2018) (Fig 4A–D and Tables S4 and S7), suggesting loss of PRDM16 triggers a shift away from ventricular working CM fate. Also at the chromatin level, and in line with the transcriptomic remodeling, most genes related to more open DARs were part of the atrial or conduction signature, whereas most genes related to closed DARs were part of the ventricular working CM gene signature (Fig S6C and D and Tables S5 and S8). The expression pattern of several atrial, ventricular, working, or conduction genes was validated by RT–qPCR or at the protein level by immunostaining or immunoblotting (Fig S7A–D).

The remarkable shift toward a VCS signature was in part due to an increase in the proportion of PFs. Indeed, when increasing the resolution of CM clustering, we identified a small CM subpopulation (cluster 7) with a significantly higher expression of PF genes (e.g., *Cntn2*, *Cpne5*), which are also significantly increased in *Prdm16^cKO* CMs compared with *Prdm16^WT* cells (Figs 4E and S8A and Table S3). The proportion of this PF cluster increased ~5-fold from 0.5% in *Prdm16^WT* CMs to 2.3% in PRDM16-deficient CMs (Fig S8A and B and Table S3). The increase in PFs was confirmed at the protein level by the significantly (~5.4-fold) increased staining for the specific mature PF marker CONTACTIN-2 (Fig 4F) (Pallante et al, 2010) and by the apparent increase in the RNA expression of additional PF markers (Fig S8C). Altogether, PRDM16 loss during development caused expression changes in CMs reflecting alterations related to a shift toward both atrial and conduction cell fates, the latter resulting in a hyperplastic (distal) VCS.

### PRDM16 suppresses the activity of master regulators of atrial and conduction fates

To better understand how PRDM16 regulates gene expression in CMs, we performed TF motif enrichment analysis, focusing on open DARs in *Prdm16^cKO* CMs. Using both chromVAR and HOMER, we revealed 309 and 59 significantly enriched motifs, respectively, of which 31 overlapped. Some of these motifs are known to be associated with PRDM16, including TGIF1/2, TEAD1-4, and MEF2C (Fig S9A). Furthermore, several other motifs represent members of the GATA and TBX families, known to be involved in CM cell fate decision-making (Fig S9A). When performing the same motif analysis on the subset of open atrial or conduction DARs (Table S8), the combined chromVAR and HOMER analysis revealed similar motifs as described above (Fig 5A and B). Remarkably, for both atrial and conduction genes, TBX5 motifs were present in about 50% of the DARs, in line with its known role as a master regulator of atrial and conduction fates (Fig 5A and B) (Pawlak et al, 2019; Choquet et al, 2021; Cao et al, 2023; Sweat et al, 2023). Another commonly

present pathway known to be suppressed by PRDM16 was the TGFβ pathway, with TGIF motifs in 50% of DARs associated with atrial genes and SMAD2-3 motifs in 7% of DARs associated with conduction genes (Fig 5A and B). Furthermore, PRDM16 loss also significantly increased the expression of some of these TFs (i.e., *Tbx5*, *Mef2c*, *Smad2*, *Pbx3*), suggesting that PRDM16 puts a double brake on their activity by (1) blocking the accessibility of DNA binding sites of these TFs and (2) by regulating the expression of these TFs (Fig 5C).

Knowing that certain master regulators were differentially expressed after PRDM16 loss, and after identifying DEGs of atrial and VCS cell fates, we calculated TF–gene associations and constructed gene regulatory networks (GRNs) for both atrial and conduction cell fates. Within each fate-specific GRN, we then looked at how PRDM16 could regulate such a network. To construct these GRNs, we used the R package Functional Inference of Gene Regulation (FigR), which uses our RNA count data and paired ATAC peak counts. FigR first identifies significant peak–gene associations, linking *cis*-regulatory elements to target genes, and defines these as domains of regulatory chromatin (DORCs). We identified 8,221 DORCs and filtered the associated genes based on our previously identified atrial and conduction DEGs (to construct an atrial- and conduction-specific network; Table S7), retaining the DORCs of 165 atrial and 162 conduction coding genes, respectively. FigR then identifies TF modulators associated with these domains to construct a GRN. A regulation score is calculated for each TF-DORC association. Averaging these regulation scores for each TF identifies them as activators (positive mean regulation score) or repressors (negative mean regulation score) (Fig 5D and E and Table S9). Some of the top five activating TFs for atrial and conduction genes that emerged from our FigR analysis were common with those extracted from the motif analysis, that is, MEF2C and TBX5 (Fig 5D and E and Table S9). Plotting a heatmap based on the regulation score of the top five activator TFs and associated DORCs of the genes of our atrial and conduction GRNs alongside PRDM16 clearly revealed that PRDM16 was opposing the activity of these master activators (Fig 5D and E). Our GRN analysis showed that only a minority of the atrial and conduction DEGs were uniquely regulated by PRDM16 (Fig S9B and C). Together, this indicates that PRDM16 represses atrial and conduction cell fates in the heart primarily by confining the activity of master regulator TFs.

### PRDM16 orchestrates CM fate decision by acting on promoters and distant enhancers

We then focused on these peak–gene associations to study more precisely how PRDM16 regulates genes related to atrial and conduction fates. We combined the 165 atrial and 162 conduction DEGs from our GRNs, and focused on those genes with significant DARs. This resulted in a compiled list of 120 significant DARs associated

---

representative for hypertrophy (*left*), fatty acid (FA) metabolism (*middle*), or TGFβ signaling (*right*). The absolute number and proportions of DEGs for each term are indicated. **(D)** Bar graph showing the numbers of differentially accessible regions (DARs) identified in CMs. Green and purple bars represent the number of more and less accessible regions, respectively. Pie charts represent the annotation (expressed in %) of more open (*top*) or more closed (*bottom*) DARs in CMs. UTR, untranslated region; TTS, transcription termination site. **(E)** Functional annotation on DEGs higher expressed in *cKO* CMs showing the top 25 cardiac-related terms. Terms related to chamber type, conduction, and growth are highlighted in red.

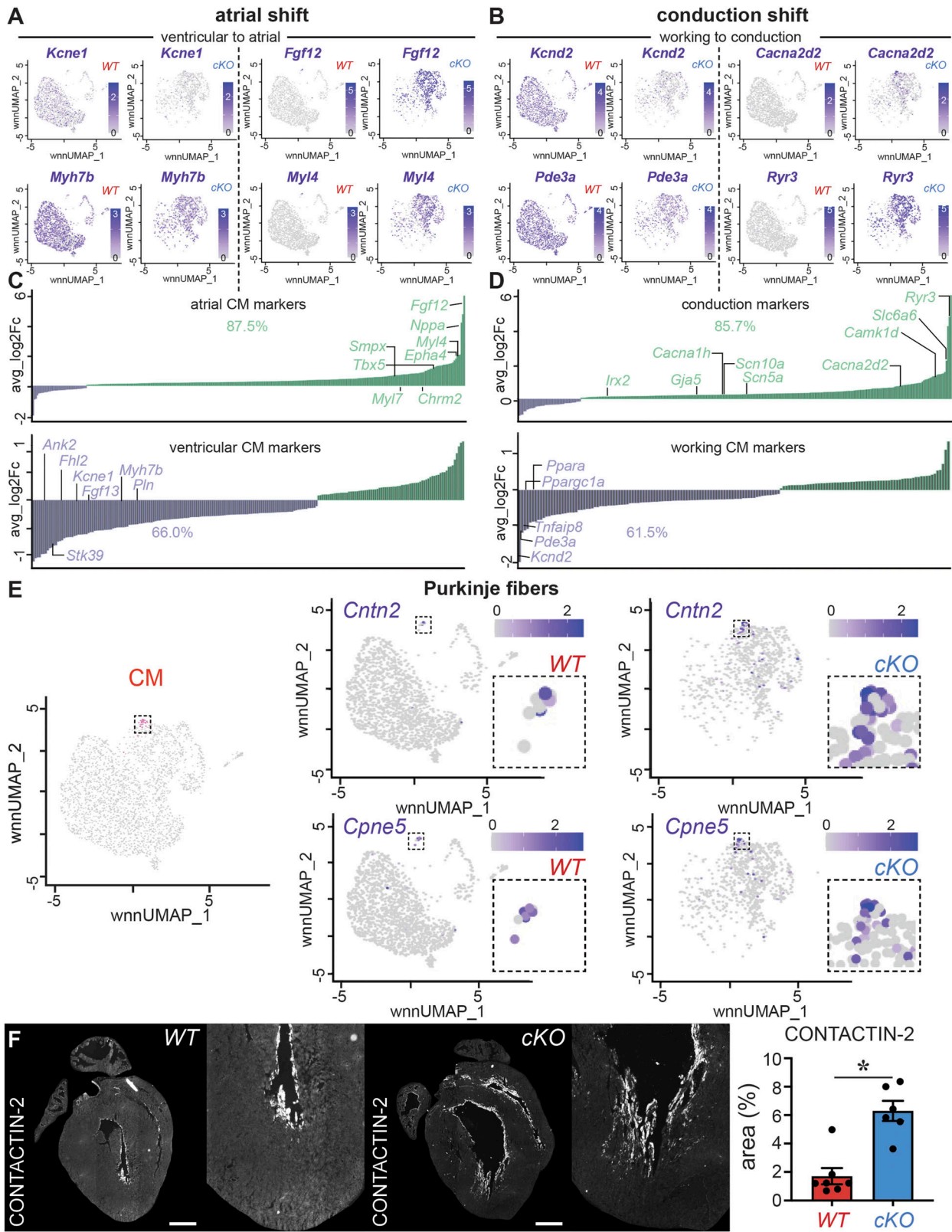

**Figure 4. PRDM16 loss in CMs causes a shift toward atrial and conduction fates.**
**(A)** Integrated Uniform Manifold Approximation and Projections (UMAPs) showing the enriched expression of ventricular genes (*Kcne1*, *Myh7b*; *left*) in *WT* cardiomyocytes (CMs) and the enriched expression of atrial genes (*Fgf12*, *Myl4*; *right*) in *Prdm16*-deficient (*cKO*) CMs. The color scale represents log(fold change) from low (white) to high (purple). **(B)** Integrated UMAPs showing the enriched expression of ventricular working CM genes (*Kcnd2*, *Pde3a*; *left*) in *WT* CMs and the enriched expression

with 67 coding DEGs. First, we looked at whether PRDM16 DNA binding motifs, as identified in the CIS-BP database (Weirauch et al, 2014), were present in these DARs. We found such motif(s) in DARs associated with 23 out of 67 DEGs (Fig 6A). We then identified PRDM16 binding motifs present in the promoter region (+/– 2 kb of the transcriptional start site [TSS]) and cross-checked these regions with a previously published PRDM16 ChIPseq dataset (Wu et al, 2022) generated on CMs from E13.5 hearts. This revealed that binding to the same promoter region was confirmed for 5 DEGs (e.g., *Ryr3*, *Lrp8*; Fig 6A, **green box**; Fig 6B), suggesting that PRDM16 can directly bind to the promoter of these genes. On the contrary, the PRDM16 ChIPseq dataset also revealed potential binding to the promoter region of 5 DEGs (e.g., *Nppa*, *Gpx3*) in the absence of a PRDM16 binding motif, suggesting indirect binding of PRDM16 (Fig 6A, **gray box**; Fig 6C). For the regions outside the promoters, we cross-checked with the ENCODE database on P0 hearts to identify which of these corresponded to enhancer regions (Fig 6A, **pink boxes**). This revealed 20 DEGs with enhancers, the majority of which were marked with the active enhancer mark H3K27Ac (e.g., *Sparc*, *Camk1d*; Fig 6A, **pink boxes**). In four cases (e.g., *Slc6a6*, *Tbx5*), the enhancer regions were marked with the mark H3K4me3 (Fig 6A, **pink boxes**). Remarkably, ChIPseq-validated TBX5 binding sites (Steimle JD et al, 2018; Akerberg et al, 2019) lay in close vicinity for 4 out of 5 genes that harbored a confirmed direct binding site for PRDM16 and for all five genes with suggested indirect PRDM16 binding in their promoter region (Fig 6A, **gray** and **green boxes**; Fig 6B and C). This suggests both TFs may co-occupy regulatory regions of the same genes. Finally, to link distal cis-regulatory elements to the TSS of target genes, we performed a peak-to-gene analysis. For seven atrial or conduction markers (including *Tbx5* and *Slc6a6*), we identified such distant enhancers, some of which harbored a PRDM16 binding motif and/or a ChIPseq-validated TBX5 binding site (Fig 6D). Thus, PRDM16 favors the ventricular CM cell fate by modulating atrial and conduction gene expression, which involves both direct and indirect binding to target regions (both promoter and distant enhancer regions). Moreover, a number of these target regions are also recognized by TF TBX5, which is known to favor these alternative cell fates.

## Discussion

Cardiac cells are a heterogeneous mixture of subtypes each responsible for different functions to ensure continuous cardiac contraction. To establish these different cellular subtypes, appropriate lineage decisions must be taken and maintained. TFs play an orchestrating role in the decision-making process from early on during cardiac development. Here, we revealed that PRDM16 is one of these key TFs that co-regulates the proper specification of ventricular working CMs mainly by suppression of genes typical for atrial and conduction fates, by opposing the activity of master regulators. PRDM16 loss in CMs during development resulted in their differentiation toward more atrial- and conduction-like cells, thereby causing (distal) VCS hyperplasia (as graphically summarized in the graphical abstract). The latter is likely co-responsible for the premature death of a large proportion of PRDM16-deficient pups.

Arndt et al (2013) showed that *PRDM16* mutations in humans and *prdm16* knockdown in zebrafish are associated with cardiomyopathy potentially because of cell-autonomous changes in CMs. Prompted by these observations, studies in mice were performed to investigate the CM-specific role of PRDM16 using the Cre-lox system (Arndt et al, 2013; Cibi et al, 2020; Nam et al, 2020; Wu et al, 2022; Kramer et al, 2023). In these studies, different Cre drivers were used and the phenotypic consequences of Cre-driven *Prdm16* deletion were remarkably diverse, because of the distinct cell populations being targeted and the specific developmental time frames during which Cre recombinase was active. For our studies, we used the Sm22α-Cre line, which is active in the developing heart, and faithfully deleted *Prdm16* between E8.5 and E12, a crucial time window during cardiac development. Recently, sexual dimorphism in cardiomyopathy patients because of *PRDM16* mutations has been reported, with females having a higher chance of developing cardiomyopathy (Kramer et al, 2023). In mice, this sexual dimorphism was also seen in models that did report a mild phenotype—that is, without lethality—after *Prdm16* deletion (Nam et al, 2020; Kramer et al, 2023; Kuhnisch et al, 2024). Here, we did observe a more dramatic phenotype (including early mortality) but found no sex differences. This may be related to the fact that the severe phenotype may overrule the sex effect of PRDM16 loss in CMs.

One of the main phenotypic traits already present at P7 and persisting until adulthood in *Prdm16*^*cKO* mice was a prolonged QRS complex, which indicates a delayed action potential. At least three observations from our study may be responsible for the abnormal ECG. First, we noted the significantly reduced expression of ion channel genes implicated in cardiac electrophysiology, including *Kcnd2* and *Fgf12*. Mutations in *Kcnd2* have been associated with long-QT syndrome (Berger et al, 2010; Marks, 2013; Haissaguerre et al, 2016) and/or sudden cardiac death (Toib et al, 2017; Farrell et al, 2018). Moreover, mice carrying a gain-of-function missense mutation in *FGF12* were recently shown to suffer from bradycardia

of ventricular conduction genes (*Cacna2d2*, *Ryr3*; *right*) in *cKO* CMs. The color scale represents log(fold change) from low (white) to high (purple). **(C)** Bar graphs show the expression of atrial (*top*) or ventricular (*bottom*) markers (Cao et al, 2023) overlapping with the differentially expressed gene lists and their average Log$_2$FC, revealing the higher expression (green; average Log$_2$FC > 0) of 87.5% of the atrial genes and the lower expression (purple; average Log$_2$FC < 0) of 66.0% of the ventricular genes. The full marker lists are shown in Table S7. **(D)** Bar graphs show the expression of ventricular conduction (*top*) or working (*bottom*) markers (Shekhar et al, 2018) overlapping with the differentially expressed gene lists and their average Log$_2$FC, revealing the higher expression (green; average Log$_2$FC > 0) of 85.3% of the ventricular conduction genes and the lower expression (purple; average Log$_2$FC < 0) of 61.5% of the ventricular working genes. The full marker lists are shown in Table S7. **(E)** Integrated UMAP of re-clustered CMs subclustered at high resolution highlighting the PF cluster in pink (*left*). The complete subcluster analysis is shown in Fig S8. The middle top panel (*WT*) and top right panel (*cKO*) show the expression of the PF marker CONTACTIN-2 (*Cntn2*) in CMs. The middle lower panel (*WT*) and lower right panel (*cKO*) show the expression of the PF marker *Copine5* (*Cpne5*) in CMs. Insets focus on the Purkinje cluster. The color scale represents log(fold change) from low (white) to high (purple). **(F)** Representative images of CONTACTIN-2 protein staining identifying the PFs in *WT* versus *cKO* P7 mouse hearts. The bar graph shows quantitative analysis of relative CONTACTIN-2 area as a % of the whole ventricles (*n* = 7/6). Quantitative data are expressed as the mean ± SEM; *$P < 0.05$ by a *t* test. Scale bars: 500 *μ*m (F).

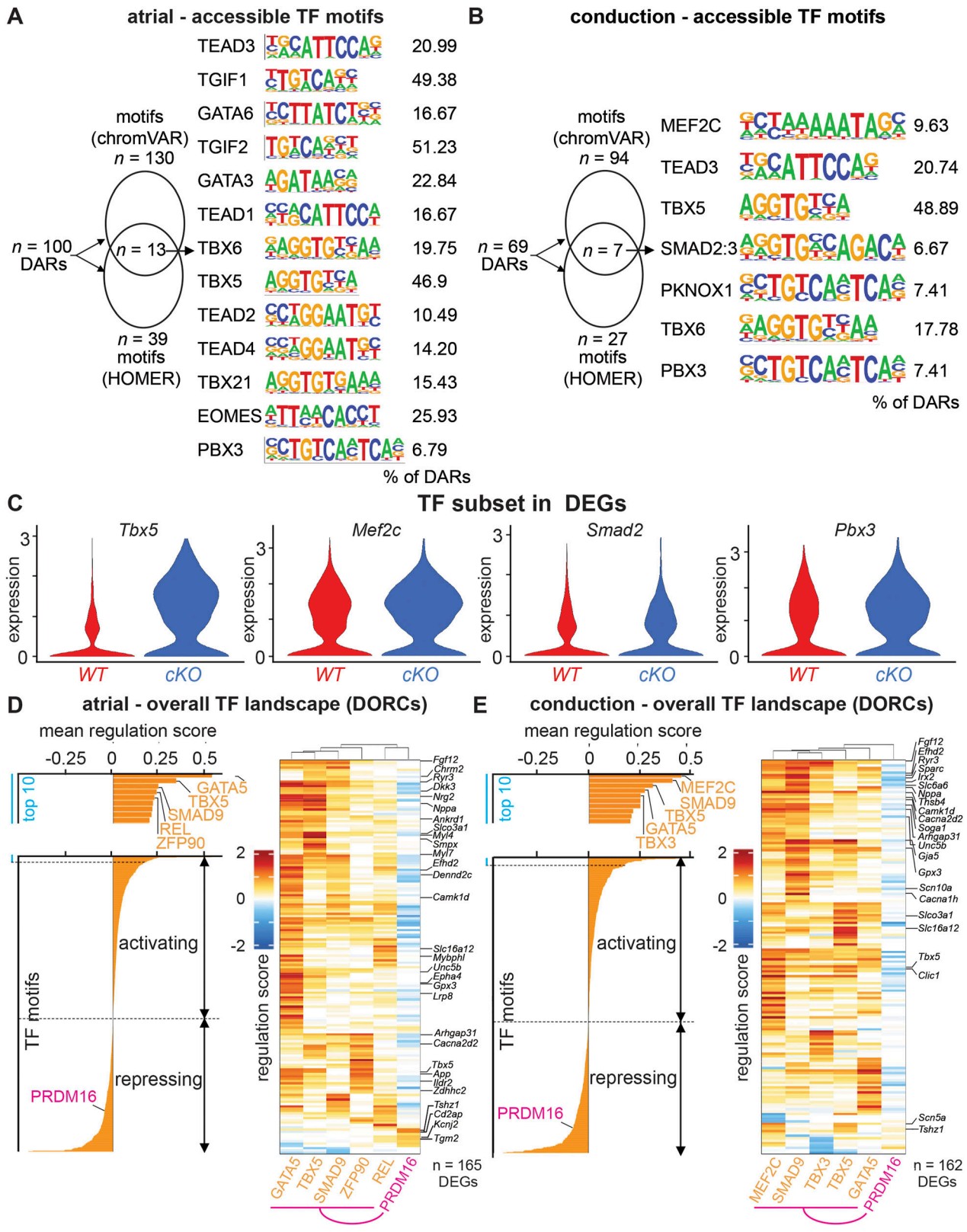

**Figure 5.  PRDM16 suppresses the activity of master regulators of atrial and conduction fates.**
**(A)** Scheme showing 100 atrial-specific more open (logFC > 0) differentially accessible regions (DARs) in *Prdm16*-deficient (*cKO*) versus *WT* cardiomyocytes (CMs) that were scanned for transcription factor (TF) motifs revealing 130 and 39 motifs, using chromVAR and HOMER, respectively, of which 13 overlapped. Overlapping TF motifs are listed on the right along with the percentage of DARs containing the motif sequence. **(B)** Scheme showing 69 ventricular conduction-specific more open (logFC > 0) DARs in

and sudden cardiac death (Veliskova et al, 2021). Second, the increased fibrosis seen in PRDM16-deficient myocardium may cause the formation of heterotypic junctions between myofibroblasts and CMs resulting in lower electrical conductivity (Rubart et al, 2018). Finally, structural abnormalities in the VCS are invariably associated with arrhythmias (Choquet et al, 2021). In the majority of reported cases (e.g., deficiency of ETV1, NKX2.5, NCAM-1), the abnormalities represent VCS hypoplasia, but few cases of hyperplasia have been reported to date (Shekhar et al, 2018; Choquet et al, 2021). Only recently, a *Hand1* mutant lacking an LV-specific enhancer was reported to cause VCS hyperplasia with reduced conduction velocity (Vincentz et al, 2019). Similarly, we found (distal) VCS hyperplasia in *Prdm16^cKO* hearts. Wu et al showed that during early development, PRDM16 cooperates with HAND1 regarding LV specification (Wu et al, 2022). However, *Hand1* expression was not changed in our setting because of the different time points in cardiac development in both studies. The hyperplasia of the VCS in our PRDM16-deficient mouse model was manifest at the histological level by an increase in the distal region consisting of PFs. The observed prolonged QRS is indeed compatible with defects in the distal VCS (Christoffels & Moorman, 2009; Choquet et al, 2020). By subclustering, we were also able to resolve a cell population compatible with the PF gene signature. The transcriptional changes we observed revealed an increase in several PF markers including *Cntn2* and *Ryr3*. PFs have specific electrophysiological properties and particular $Ca^{2+}$ dynamics with RYR3 being expressed about 100-fold higher in PFs versus working myocardium and being more sensitive to $Ca^{2+}$ than RYR2 to stabilize the electric function of PFs (Pallante et al, 2010; Haissaguerre et al, 2016; Daniels et al, 2017). At the same time, the high sensitivity to $Ca^{2+}$ makes PFs more arrhythmogenic, which may result in sudden cardiac death because of ventricular fibrillation (Haissaguerre et al, 2016). As no less than 60% of the *Prdm16^cKO* pups died before weaning age, it is tempting to speculate that the aberrant VCS in these mice contributes to arrhythmias that could lead to premature death. Although this pattern is consistent with such a cause of death, further analysis is required to confirm this.

It has been shown that the VCS in part differentiates from trabecular progenitor CMs because of a slower proliferation rate (Choquet et al, 2021). Interestingly, during embryonic development these trabeculae disappear because of an even faster proliferating compact myocardium, whereas the VCS continues to recruit cells and continues to mature postnatally (Choquet et al, 2020). Wu et al documented a role of PRDM16 in determining the compact ventricular CM fate at the expense of trabecular gene expression at E13.5, a time point in line with the disappearance of the trabeculae (Fig S1A) (Wu et al, 2022). We found that taking a transcriptomic

snapshot of cardiac *Prdm16* deletion at P7 (also falling within the temporal window of VCS development; Fig S1A) reflected the role of PRDM16 in the decision to become part of the ventricular working myocardium, which requires the suppression of conduction marker genes. Hence, our study complements the findings of Wu et al (2022) on the detrimental role of PRDM16 operating as an important regulator of compact versus trabecular and working versus conduction CMs. When using a previously published reference signature for trabecular CMs (Li et al, 2016), we noticed only a partial and rather mild transcriptomic shift toward a trabecular fate upon PRDM16 loss when comparing to Wu et al (2022), likely because of the later time point in our study (Fig S10 and Table S4). However, we did see clear anatomic signs of hypertrabeculation, in accordance with previous reports demonstrating hypertrabeculation may persist until adulthood, whereas related expression changes in CMs subside by P3 (Luxan et al, 2013). Vice versa, besides the focus by Wu et al on trabeculation, they also described up-regulation of a neuron-like transcriptomic signature upon PRDM16 loss in E13.5 embryos but less dramatic compared with our data because of the earlier transcriptomic snapshot in their study (Wu et al, 2022). This again supports the differentiation shift during the embryonic development of CM progenitors toward a conduction identity upon PRDM16 loss. In addition, Nam et al (2020) showed that the transcriptome of 1-mo-old PRDM16-deficient hearts with prolonged QRS also displayed the increased expression of conduction marker genes in CMs, including *Cntn2, Ryr3*, and *Cacna2d2*. Hence, PRDM16 still determines the cardiomyocyte cell fate postnatally, during the late phase of VCS development (Nam et al, 2020).

Similar to other studies (Cibi et al, 2020), our multiomics analysis revealed that the primary function of PRDM16 is to suppress alternative cell fates. We therefore pinpointed our mechanistic studies toward how it elicited this effect by primarily focusing on up-regulated genes after PRDM16 loss. Nevertheless, there was also a significant set of genes that was down-regulated upon PRDM16 loss, including many genes typical for ventricular working CMs. This dual activity of PRDM16 has been reported in other cell types, including ECs, SMCs, neurons, and adipocytes (Kajimura et al, 2008; Aranguren et al, 2013; Baizabal et al, 2018; Wang et al, 2023). Our TF motif and GRN analyses commonly revealed that PRDM16 most likely mediates its repressive effects on atrial and conduction marker genes through opposition of master regulator TFs, most notably TBX5. Not only was TBX5 up-regulated, but we also noticed increased enrichment for TBX5 binding motifs in peaks associated with up-regulated genes. Intriguingly, a similar motif analysis by Wu et al showed increased enrichment for TBX5 binding motifs at PRDM16 peaks of down-regulated genes after ChIPseq of *Prdm16^cKO* versus *Prdm16^WT* embryonic LVs. This suggested a positive

---

*cKO* versus *WT* CMs that were scanned for TF motifs revealing 94 and 27 motifs, using chromVAR and HOMER, respectively, of which seven overlapped. Overlapping TF motifs are listed on the right along with the percentage of DARs containing the motif sequence. **(C)** Violin plots of enriched TFs differentially expressed in *cKO* versus *WT* CMs. Gene regulatory network (GRN) analysis using FigR identifying domains of regulatory chromatin (DORCs) and associated genes. **(D, E)** To obtain atrial-specific (D) or conduction-specific (E) GRNs, genes were filtered using established atrial (Cao et al, 2023) or conduction (Shekhar et al, 2018) gene signatures and for being differentially expressed. Bar graphs represent the mean regulation score (y-axis, $\log_{10}$-transformed per TF across all DORCs). The upper section of the bar graphs zooms in on the top 10 "master activator" TFs (positive mean regulation score) of the GRN. Heatmaps represent TF-DORC associations colored according to their regulation score (blue = negative or repressor; red = positive or activator), with DORCs representing the associated atrial or conduction differentially expressed genes (y-axis). TFs (x-axis) represent the top five master activators alongside PRDM16. The purple line indicates the repressing regulation by PRDM16 compared with the other TFs. The raw data for motif and heatmap analysis are included in Table S9. The color scale represents the regulation score per TF-DORC association.

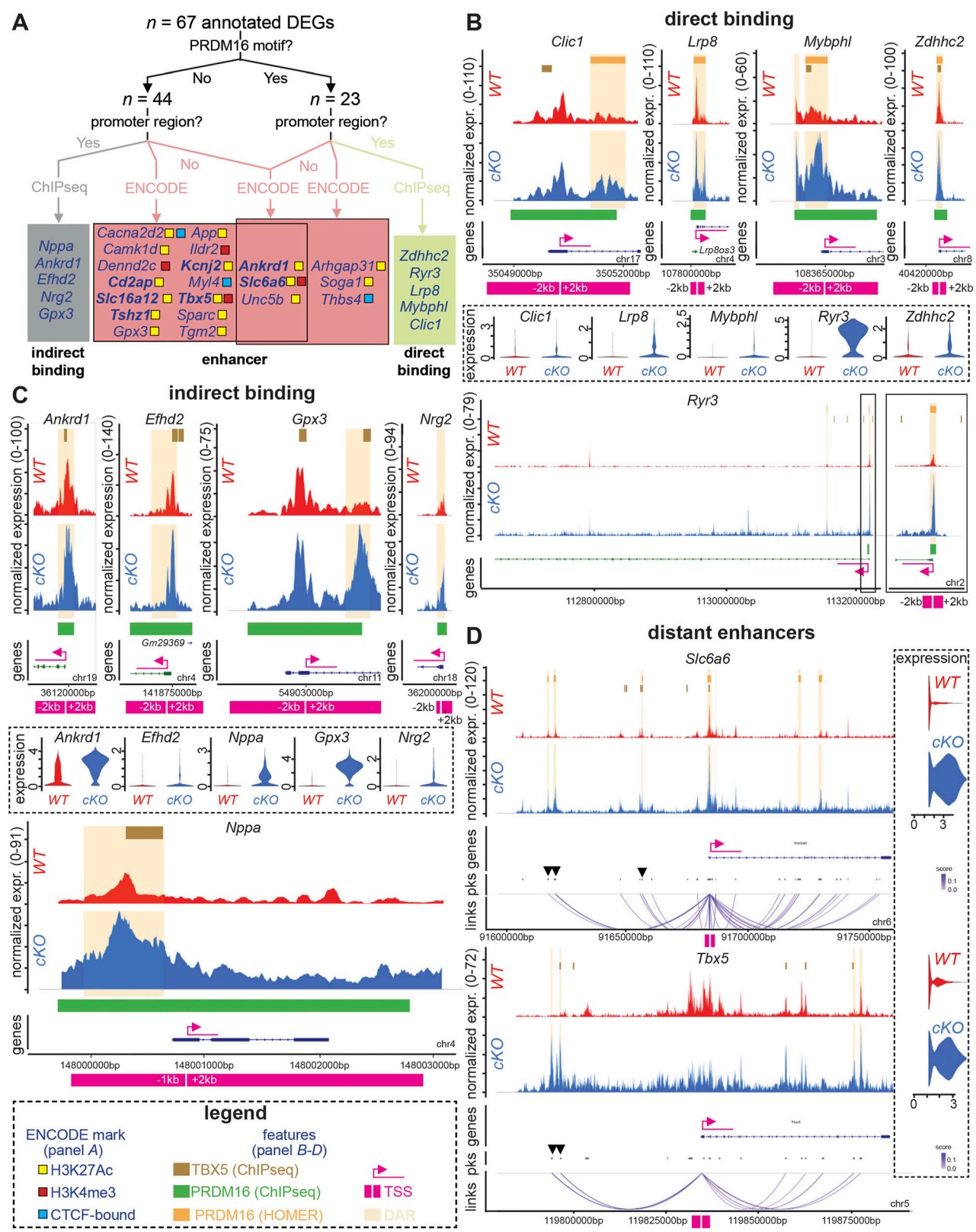

**Figure 6. PRDM16 orchestrates CM fate decision by acting on promoters and distant enhancers.**
**(A)** Decision tree identifying PRDM16 targets in differentially accessible regions associated with 67 atrial- and/or conduction-specific coding differentially expressed genes (DEGs) and located inside or outside the promoter region. A combination of HOMER-PRDM16 binding motif analysis, publicly available PRDM16 ChIPseq (E13.5 cardiomyocytes), and ENCODE datasets (P0 hearts) was applied to determine whether PRDM16 directly or indirectly binds their promoter (green and gray boxes,

cooperation between PRDM16 and TBX5 in the decision between compact versus trabecular CMs rather than an opposition as we propose here in conduction versus working CMs (Wu et al, 2022). Interestingly, a recent study put forward TBX5 as a master regulator of the atrial fate that was also important for maintaining this fate later in life (Sweat et al, 2023). Whether PRDM16 also maintains ventricular (working) CM fate by cooperation with positive regulators/facilitators of this fate, for example, ERRα/γ or HEY2, remains to be determined (Koibuchi & Chin, 2007; Sweat et al, 2023).

Although previous studies suggested that PRDM16 mainly acts through indirect association with DNA (Baizabal et al, 2018; Wu et al, 2022), our analysis revealed a multitude of PRDM16 binding sites in the promoter region of atrial and conduction genes, a fraction of which was validated by crossover with the ChIPseq dataset of Wu et al (2022). At the same time, overlapping the atrial and conduction DEG lists with the same ChIPseq dataset also revealed PRDM16 binding in their promoters in the absence of the binding motif, corroborating that PRDM16 may also associate with DNA as part of a transcriptional complex, in line with previous studies. A noteworthy limitation to this crossover analysis is the already abovementioned temporal mismatch between the study of Wu et al (2022) and our study (Fig S1A). Unlike previous studies, our analysis also included an evaluation of the effects of PRDM16 deficiency on chromatin accessibility, inspired by previous observations that describe epigenetic actions by PRDM16 either through association with histone-modifying enzymes or by its own methylation activities (Pinheiro et al, 2012; Di Zazzo et al, 2013; Harms et al, 2014; Li et al, 2015; Baizabal et al, 2018; Cibi et al, 2020; Jiang et al, 2022). ATACseq revealed that deletion of Prdm16 in CMs resulted in a significant number of DARs, the majority of which were associated with increased accessibility, further supporting the major repressive effect of PRDM16. Integration of RNAseq and ATACseq enabled us not only to perform TF motif and GRN analyses, but also to address the question whether PRDM16 acts on distant cis-regulatory elements, a mechanism that has been detected earlier during PRDM16-mediated neuronal differentiation (Baizabal et al, 2018). The involvement of distant enhancers was indeed shown for a subset of genes, including Tbx5. Altogether, the involvement of epigenetic activity governed by PRDM16 adds another layer of complexity to its mode of action in CMs.

In this study, we showed that PRDM16 orchestrates the decision for a CM (progenitor) to differentiate toward a mature ventricular CM. Therefore, it will be very appealing to test whether the overexpression of PRDM16 is sufficient to induce such a fate in induced

pluripotency stem cell–derived CMs (Kodo et al, 2016; Funakoshi et al, 2021). If successful, this may represent an inexhaustible source of mature ventricular CMs to be used for regenerative therapies in post-infarct patients for seeding regenerative patches of myocardium, which in their current status lack proper mature ventricular physiology.

# Materials and Methods

## Animal strains and husbandry

Animal experiments were approved by the KU Leuven Animal Ethics Committee and performed under the Committee's guidelines. Animals were housed under specific pathogen-free conditions and kept in a 12/12-h light/dark cycle and allowed access to standard rodent chow and water ad libitum. Sex was not determined for pups found dead in their cage. Three genetically modified mouse strains were used (Fig S2A): (1) ubiquitous constitutive Prdm16 knockout mice (Prdm16 Gt(OST67423)Lex; referred to as "Prdm16-LacZ") purchased from the Mutant Mouse Resource & Research Center (MMRRC) and backcrossed for nine generations on a C57BL/6 background (Kajimura et al, 2008); (2) conditional Prdm16 knockout mice generated by inter-crossing homozygous Prdm16^lox/lox mice (available through B. Spiegelman, Boston, USA; on a C57BL/6 background [Cohen et al, 2014]) with LoxP sites flanking exon 9 of the Prdm16 gene and Sm22α-Cre driver mice (available through J. Herz, Texas, USA; on a mixed CD1/C57BL/6 background [Holtwick et al, 2002]), which we refer to as "Prdm16^cKO" (and their corresponding Cre-negative littermates "Prdm16^WT"); (3) a Cre activity reporter strain generated by inter-crossing Sm22α-Cre driver mice with mice harboring a R26R CAG-boosted eGFP (RCE) cassette with a floxed STOP codon before the GFP-encoding gene under the CAG promoter in the Rosa26 locus (Sousa et al, 2009). For genotyping, genomic DNA was extracted using tissue from mouse ears and put overnight in lysis buffer. Isopropanol-based isolation of the DNA was performed, and alleles of interest were genotyped using Thermo Fisher Scientific PuReTaq PCR beads. Primers are listed in Table 2.

## Tissue/embryo harvesting

When euthanized for tissue collection, adult animals were injected with Dolethal (66.7 μg/g), whereas 7-d-old pups were decapitated. The chest was cut open, and organs were dissected out and snap-

respectively) or potentially interacts with their enhancer regions (pink boxes; genes in bold-face font are those associated with distant enhancers). ChIP, chromatin immunoprecipitation. The color code of enhancer regions corresponds to the one used by ENCODE: H3K27Ac: histone-3-lysine-27 acetylated (yellow), H3K4me3: histone-3-lysine-4 trimethylated (red), and CTCF-bound: CCCTC-binding factor (blue). ATAC peaks of the promoter region of WT (red) and Prdm16-deficient (cKO, blue) cardiomyocytes for genes from the green box in (A), representing cases of ChIPseq-validated direct PRDM16 binding to the promoter region. ATAC peaks of the promoter region of WT (red) and cKO (blue) cardiomyocytes for genes from the gray box in (A), representing cases of ChIPseq-validated indirect PRDM16 binding to the promoter region. (D) Peak-to-gene link plots identifying distant enhancer regions in WT (red) versus cKO (blue) cardiomyocytes in Slc6a6 and Tbx5. Purple strings indicate the peak-to-gene link; the color scale represents correlation significance (from 0 to 1). Arrowheads represent enhancers identified by the ENCODE database. Transcriptional start site is marked in (B, C, D) by fuchsia arrow and white line +/− 2-kb region indicated by fuchsia line. Differentially accessible regions are highlighted in (B, C, D) by light-orange shades. Brown lines (top) in (B, C, D) represent ChIPseq-validated (in E9.5 or E12.5 cardiomyocytes) (Steimle et al, 2018; Akerberg et al, 2019) TBX5 binding site regions. Green lines (bottom) in (B, C) represent PRDM16 target peaks validated by ChIPseq in E13.5 CMs (Wu et al, 2022). Dark-orange lines (top) in (B) represent HOMER-predicted PRDM16 binding sites. The differential expression of the genes is shown as violin plots. expr, expression; chr, chromosome; pks, peaks.

**Table 2. Detailed resources table.**

| Antibody | Source | Identifier |
|---|---|---|
| Anti-PRDM16 antibody (sheep) | R&D Systems | Cat#AF6295 (Bauters et al, 2017) |
| Anti-GFP antibody (chicken) | Abcam | Cat#ab13970 |
| Anti-LAMININ antibody (rabbit) | Sigma-Aldrich | Cat#L9393 |
| Anti-ENDOMUCIN antibody (goat) | R&D Systems | Cat#AF4666 |
| Anti-DESMIN antibody (goat) | R&D Systems | Cat#AF3844 |
| Anti-PCM1 antibody (rabbit) | Sigma-Aldrich | Cat#HPA023370 |
| Anti-CONTACTIN-2 antibody (goat) | R&D Systems | Cat#4439 |
| Anti-SMA-CY3 antibody (mouse) | Sigma-Aldrich | Cat#F3777 |
| Anti-MYL4 antibody (rabbit) | Thermo Fisher Scientific | Cat#PA5-119955 |
| Anti-FGF12 antibody (rabbit) | Abcam | Cat#ab231956 (Tian et al, 2024) |
| Anti-CACNA2D2 antibody (rabbit) | Novus Biologicals | Cat#NBP1-81501 (Zhang et al, 2017) |
| Anti-NPR3 antibody (mouse) | Santa Cruz | Cat#sc-515449 |
| Anti-FHL2 antibody (rabbit) | Thermo Fisher Scientific | Cat#21619-1-AP |
| Anti-CD31 antibody (rabbit) | Abcam | Cat#Ab28364 |
| **Primer** | **Sequence** | **Source** |
| Mouse *Gapdh* forward | 5′-ccgcatcttcttgtgtgcagt-3′ | Integrated DNA Technologies (IDT) |
| Mouse *Gapdh* reverse | 5′-gaatttgccgtgagtggagt-3′ | IDT |
| Mouse *Prdm16* forward | 5′-cagcacggtgaagccattc-3′ | IDT |
| Mouse *Prdm16* reverse | 5′-gcgtgcatccgcttgtg-3′ | IDT |
| Mouse *Nppa* forward | 5′-gcttcgggggtaggattgac-3′ | IDT |
| Mouse *Nppa* reverse | 5′-gaggcaagaccccactagac-3′ | IDT |
| Mouse *Nppb* forward | 5′-tgggctgtaacgcactgaag-3′ | IDT |
| Mouse *Nppb* reverse | 5′-acttcaaaggtggtcccaga-3′ | IDT |
| Mouse *Ryr3* forward | 5′-gacaggaccaggaacggaag-3′ | IDT |
| Mouse *Ryr3* reverse | 5′-gctccaccgtcttttctgga-3′ | IDT |
| Mouse *Tuba1b* forward | 5′-ccagatgccaagtgacaaga-3′ | IDT |
| Mouse *Tuba1b* reverse | 5′-gatctccttgccaatggtgt-3′ | IDT |
| Mouse *Kcne1* forward | 5′-cagcagagcctcgaccattt-3′ | IDT |
| Mouse *Kcne1* reverse | 5′-ctgaagctctccaggacacg-3′ | IDT |
| Mouse *Fgf12* forward | 5′-ctacaccctcttcaatctaattcc-3′ | IDT |
| Mouse *Fgf12* reverse | 5′-ttccccttcatgatttgacc-3′ | IDT |
| Mouse *Myl4* forward | 5′-ccaatggctgcatcaactatgaa-3′ | IDT |
| Mouse *Myl4* reverse | 5′-ccatgtgagtccaatactccgtaa-3′ | IDT |
| Mouse *Kcnd2* forward | 5′-ctgctcacggagacacaaaa-3′ | IDT |
| Mouse *Kcnd2* reverse | 5′-cggctgttggatagtggagt-3′ | IDT |
| Mouse *Pde3a* forward | 5′-agaatccatgccaccgatgt-3′ | IDT |
| Mouse *Pde3a* reverse | 5′-cccatgtgtccgtgtgtaaa-3′ | IDT |
| Mouse *Cacna2d2* forward | 5′-aattggtggagaaagtggca-3′ | IDT |
| Mouse *Cacna2d2* reverse | 5′-ggctttctggaaattctctgc-3′ | IDT |
| Mouse *Thbs4* forward | 5′-cagacagagatggcattggagac-3′ | IDT |
| Mouse *Thbs4* reverse | 5′-ggttactgacatcaggacagctg-3′ | IDT |
| Mouse *Hey2* forward | 5′-gagaagactagtgccaacagc-3′ | IDT |
| Mouse *Hey2* reverse | 5′-gcatgggcatcaaagtagcct-3′ | IDT |

| Antibody | Source | Identifier |
|---|---|---|
| Mouse *Ank2* forward | 5′-*tggaaggagcacaagagtcgt*-3′ | IDT |
| Mouse *Ank2* reverse | 5′-*cagagccagcttcactttcttg*-3′ | IDT |
| Genotyping: *Cre* allele forward | 5′-*gaccggtaatgcaggcaa*-3′ | IDT |
| Genotyping: *Cre* allele reverse | 5′-*tccaaagcatgcagagaatgt*-3′ | IDT |
| Genotyping: *floxed Prdm16* allele forward | 5′-*gagctaggcagggacactgct*-3′ | IDT |
| Genotyping: *floxed Prdm16* allele reverse | 5′-*ccagtatcagagaggcaagaa*-3′ | IDT |
| Genotyping: *Prdm16GT(OST67423)LEX* 1 | 5′-*acaggcgaggaactgtatgaaagg*-3′ | IDT |
| Genotyping: *Prdm16GT(OST67423)LEX* 2 | 5′-*ccatctgaggtcgtctgaaactgg*-3′ | IDT |
| Genotyping: *Prdm16GT(OST67423)LEX* 3 | 5′-*aaatggcgttacttaagctagcttgc*-3′ | IDT |

| Software | Source | Identifier |
|---|---|---|
| Cell Ranger Arc v1.0.1 | 10X Genomics | https://support.10xgenomics.com/single-cell-multiome-atac-gex/software/overview/welcome |
| CellBender v0.1.0 | Fleming et al (2023) | https://github.com/broadinstitute/CellBender |
| scDblFinder v1.10.0 | Germain et al (2021) | https://github.com/plger/scDblFinder |
| clusterProfiler v4.4.4 | Yu et al (2012) | https://bioconductor.org/packages/release/bioc/html/clusterProfiler.html |
| rGREAT v4.0.4 | Gu and Hubschmann (2023) | http://great.stanford.edu |
| Seurat v4.3.0 | Satija Lab | https://satijalab.org/seurat/ |
| Signac v1.10.0 | Stuart Lab | https://stuartlab.org/signac/ |
| HOMER v4.11 | Heinz et al (2010) | http://homer.ucsd.edu/homer/ |
| chromVAR v1.18 | Schep et al (2017) | https://greenleaflab.github.io/chromVAR/articles/Introduction.html |
| FigR v0.1.0 | Kartha et al (2022) | https://buenrostrolab.github.io/FigR/ |
| Fiji v2.14.0 | Schindelin et al (2012) | https://imagej.net/software/fiji/#publication |
| NIH ImageJ v1.53 | | https://imagej.net/nih-image/ |
| GraphPad Prism v9.4.1 | GraphPad Software | https://www.graphpad.com/features |
| Vevo LAB v5.5.1 | FUJIFILM VisualSonics, Inc. | https://www.visualsonics.com/product/software/vevo-lab |
| QuPath v0.4.0 | Bankhead et al (2017) | https://qupath.github.io/ |
| ZEN Microscopy Software | ZEISS Group | https://www.zeiss.com/microscopy/en/products/software/zeiss-zen.html |
| ToppGene | Chen et al (2009) | https://toppgene.cchmc.org/ |

frozen (for cryosectioning, RNA or protein isolation) or post-fixed overnight in 4% PFA for further histological processing. Timed matings were set up for embryo collection at E9.5, E10.5 E11.5, E14.5, and E17.5. Pregnant dams were killed by cervical dislocation, followed by dissection of the uterus that was put in ice-cold PBS. Then, embryos were dissected out one by one in ice-cold PBS, followed by overnight fixation in PFA at 4°C for further histological processing, or embedded in Tissue-Tek and snap-frozen for cryosectioning.

## Echocardiography

Echocardiography was performed on both pups and adult animals using a Vevo2100 or Vevo3100 system (FUJIFILM VisualSonics). Pups were subjected to echocardiography at P7. Pups were anesthetized briefly by 1.5–2% isoflurane in 2.5% $O_2$, and immediately fixed on their back on a preheated heating pad. ECG gel was used to make

the connection between the paws and the ECG pads. Preheated echo gel was applied prior to obtain a short-axis view (SAX) image. Pups were put back with the mother immediately after recording. Adult animals were put in an anesthetic induction chamber, and anesthesia was induced with 5% isoflurane in 2.5% $O_2$. Once asleep, isoflurane levels were reduced to 1.5–2% and the animal was placed on its back on a heating plate containing ECG pads. After applying ECG gel on both fore and hind paws, they were fixed on the ECG pads to monitor ECG, respiratory rate, and heart rate of the mice during the procedure. A rectal probe was used to monitor body temperature, and an extra heating lamp was used to keep body temperature stable at 37°C. Once body temperature, heart rate, and breathing were stable (i.e., 37°C, ±500 beats per minute, and ±100 respiratory rate, respectively), echo recording was performed. The following images of the heart were captured: SAX, M-mode through SAX, long-axis view (LAX), and apical four-chamber view. Flow rate

through the pulmonary artery (via SAX view) and mitral valve (via apical four-chamber view) was measured using pulsed-wave Doppler where the degree of the Doppler angle was kept equal for each animal. Mitral valve movement was assessed via tissue Doppler at the septal base of the mitral valve visualized on the apical four-chamber view. ECG data were retrieved from the surface ECG of the Vevo2100 and Vevo3100 imaging system. ECG of 25 cardiac cycles was averaged. QRS was determined manually, where Q was the minimum voltage before the R-peak, R was the maximum voltage, and S was the minimum voltage immediately after the R-peak. The time between Q and S was defined as QRS duration. Maximum amplitude was the maximum voltage of the R-peak.

## Histology and morphometry

### PRDM16 *expression*
To study the expression pattern of PRDM16, two different techniques were used. The first method took advantage of the presence of a gene trap cassette in the *Prdm16* locus encoding *β-galactosidase* in *Prdm16-LacZ* mice (Fig S2A). After dissection, embryos were submerged in fixation solution (PBS containing 0.2% glutaraldehyde and 2 mmol/liter $MgCl_2$) for 20 min at 4°C before starting the X-gal staining protocol. Briefly, embryos (E9.5 and E10.5) were washed three times with PBS for 10 min and incubated overnight at 30°C with staining solution (PBS containing 1 mg/ml X-gal (Life Technologies), 5 mmol/liter $K_3Fe(CN)_6$, 5 mmol/liter $K_4Fe(CN)_6$, and 2 mmol/liter $MgCl_2$). The next day, embryos were washed three times with PBS for 10 min and fixed with 4% PFA for 2 h at room temperature (RT). Before processing for paraffin embedding, a final PBS washing step was performed. Samples were sectioned, and cross-sections were counterstained with nuclear fast red dye. PRDM16 *WT* littermates served as negative controls. For the second method, a staining protocol was optimized for immunofluorescence analysis. Briefly, tissues were dissected out, snap-frozen in liquid $N_2$, and stored at –80°C until cryosectioning. Samples were mounted with Tissue-Tek, and 7-$\mu$m sections were made using a Leica 3050S cryostat. Sections were air-dried and fixed with 4% PFA for 10 min. Next, sections were washed with Milli-Q water, washed three times with TNT (TNB + 0.05% Tween-20), and subsequently permeabilized with PBS containing 0.1% Triton for 30 min. Non-specific protein interactions were prevented by incubating slides with blocking buffer (TNB containing 10% donkey serum) for 1 h. Slides were co-incubated overnight at 4°C with sheep anti-PRDM16 antibody and anti-PCM1 antibody (to measure recombination efficiency in P7 hearts) or anti-ENDOMUCIN antibody (to delineate trabeculae in E14.5 embryos; all primary antibodies are listed in Table 2). The next day, slides were washed three times with TNT and incubated with donkey anti-sheep IgG Alexa 488 antibody and donkey anti-rabbit IgG Alexa 568 antibody or donkey anti-goat IgG Alexa 488 for 2 h at RT. Nuclear staining was obtained by incubating slides with TO-PRO-3 iodide or Hoechst for 15 min. Finally, slides were mounted with ProLong Gold Antifade. Antibodies used for immunostaining were validated during optimization of the staining procedure during which a negative control condition was included (i.e., an identical staining procedure with exclusion of the primary antibody). In case of staining for PRDM16, the antibody was validated by loss of staining upon *Prdm16* knockout.

### SM22α-Cre *activity*
*SM22α-Cre-RCE* embryos (collected at E11.5) and P7 mouse hearts were fixed overnight in 4% PFA, dehydrated in a series of ethanol and xylene, and paraffin-embedded for sectioning. Paraffin sections were stained with an anti-GFP antibody, combined with a second primary antibody labeling–specific cardiac cell–type markers (i.e., anti-DESMIN for CMs; Table 2). Briefly, paraffin sections were deparaffinized and antigens were exposed by boiling in citrate buffer or DAKO buffer depending on the primary antibody. Next, endogenous peroxidases were inactivated using methanol containing 0.3% $H_2O_2$ for 20 min. Cell membranes were subsequently lysed using 0.1% Triton X-100 in PBS, and non-specific antibody binding was blocked using 2% BSA in TBS for 1 h, followed by incubation with primary antibodies in TNB overnight. The next day, primary antibodies were washed off and slides were incubated with matching Alexa-conjugated secondary antibodies for 2 h in TNB. Where necessary, amplification was performed using Cy3 or fluorescein tyramide (FT) kits (PerkinElmer, NEL744001KT and NEL741001KT). Briefly, slides were incubated with horseradish peroxidase–conjugated streptavidin (1:100) dissolved in TNB for 30 min, washed three times with TNT, and incubated for 8 min with A-diluent containing FT (1:50) or cyanine (Cy)3-tyramide (1:50). Slides were washed and mounted with ProLong Gold Antifade mounting media.

### SMC coating
To measure vascular SMC area of the coronary arteries, paraffin cross-sections of P7 hearts were co-stained for CD31 and αSMA. After deparaffinization and dehydration, sections were boiled in Tris–EDTA–antigen retrieval for 20 min followed by a 20-min slow cool-down. Next, endogenous peroxidases were inactivated using methanol containing 0.3% $H_2O_2$ for 20 min. Slides were washed with TBS and blocked for 1 h with TNB, followed by overnight primary antibody incubation (targeting CD31 and αSMA-Cy3; Table 2). The next day, slides were washed and incubated with secondary antibody goat anti-rabbit biotin for 45 min. A TSA biotin detection kit (NEL741001KT; PerkinElmer) was used to amplify the fluorescent signal as described above. Images were taken on a ZEISS upright microscope and saved as a ZVI file, to open in Fiji. The EC layer area was measured based on the FT signal, and the SMC layer was measured based on the Cy3 signal. Both channels were opened separately in Fiji, converted to RGB, and made binary to measure the area via the wand-tracing tool. Analyses were performed by an investigator unaware of the mouse genotype. All analyses performed on images were done in Fiji, unless mentioned otherwise.

### CM hypertrophy
To measure CM size, transversal cardiac paraffin sections were stained for LAMININ. Briefly, after deparaffinization, citrate (pH 6)-based antigen retrieval, and blocking, tissue slides were embedded overnight with primary anti-LAMININ antibody (Table 2). The next day, slides were incubated with secondary Alexa 568 antibody for 2 h, followed by TO-PRO-3 for 15 min before mounting with ProLong Gold Antifade.

### Fibrosis
Cardiac cross-sections were stained with Sirius Red to assess fibrosis. Briefly, paraffin sections were deparaffinized and incubated for 90 min in freshly prepared Sirius Red (picric acid) solution

followed by differentiation in HCl for 2 min. Sections were dehydrated and mounted with DPX. Brightfield images were recorded on a ZEISS upright microscope, and fibrosis was detected and categorized into perivascular fibrosis and interstitial fibrosis. The presence of interstitial fibrosis in P7 hearts was confirmed by analyzing the sections under polarized light. Analysis was performed by an investigator unaware of the mouse genotype. Representative images for each group represent the group average.

### Compact layer expansion

P7 hearts were sectioned sagittally, and endocardium was identified via Natriuretic Peptide Receptor 3 (NPR3) or ENDOMUCIN staining to distinguish compact from non-compact myocardium. Briefly, after deparaffinization, DAKO–antigen retrieval, and blocking, tissue slides were embedded overnight with primary anti-ENDOMUCIN antibody (Table 2). The next day, slides were incubated with secondary Alexa 488 antibody for 2 h, followed by TO-PRO-3 for 15 min before mounting with ProLong Gold Antifade. For NPR3, after deparaffinization and citrate (pH 6)–antigen retrieval, endogenous peroxidases were inactivated using methanol containing 0.3% $H_2O_2$ for 20 min. Slides were washed with TBS, incubated in 0.1% Triton for 30 min, and blocked for 1 h with TNB supplemented with 10% goat serum, followed by overnight primary antibody incubation targeting NPR3 (Table 2). The next day, slides were washed and incubated with secondary antibody goat anti-mouse biotin for 45 min. A TSA biotin detection kit (NEL741001KT; PerkinElmer) was used to amplify the fluorescent signal. The width of the compact myocardium (spanning from the outer epicardium to the border of the non-compact myocardium, the latter determined based on the NPR3- or ENDOMUCIN-lined trabecular invaginations) was measured at two locations in the LV from the base to the apex. Both at the base and at the apex level, the shortest distance from the endocardium to the epicardium was used for measurement.

### VCS

To identify the (distal) VCS, frozen P7 hearts were sectioned sagittally and stained for CONTACTIN-2, a PF marker (Pallante et al, 2010). Briefly, cryosections were fixed with 4% PFA for at least 10 min. Slides were incubated with 0.1% Triton X-100 for 30 min followed by a blocking step (10% PIG in TNB, for 1 h) before overnight incubation with primary antibody against CONTACTIN-2 (Table 2). The next day, slides were incubated with secondary Alexa 488 antibody for 2 h, before mounting with ProLong Gold Antifade. Whole sections were imaged using mosaic scanning on a ZEISS upright widefield microscope. The total CONTACTIN-2 area of both ventricles was measured and normalized to the total ventricular tissue area using QuPath (https://qupath.github.io/).

### Cardiac MYL4 or FHL2 expression

P7 hearts were sectioned sagittally (MYL4) or transversely (FHL2), and atrial marker MYOSIN LIGHT CHAIN-4 (MYL4) and ventricular marker Four And A Half LIM Domains 2 (FHL2) were used to show the ventricular-to-atrial identity shift in ventricular CMs at the protein level. Briefly, after deparaffinization, Tris–EDTA–antigen retrieval, and blocking (including elimination of endogenous peroxidase activity using $H_2O_2$ in methanol), tissue slides were embedded overnight with primary anti-MYL4 antibody or anti-FHL2 antibody

(Table 2). The next day, for MYL4 detection, slides were incubated with secondary horseradish peroxidase (HRP)–labeled antibody for 45 min, followed by $DAB/H_2O_2$ developing solution. After termination of $DAB/H_2O_2$-HRP reaction, the tissue was counterstained with hematoxylin. To detect FHL2, slides were incubated with secondary Alexa 568 antibody for 2 h, followed by Hoechst staining for 15 min before mounting with ProLong Gold Antifade. MYL4 expression was quantified by measuring brown-colored area over the total myocardial area using QuPath (https://qupath.github.io/), and FHL2 expression was determined by measuring the mean fluorescence intensity signal in the heart using Fiji.

### Reverse transcription–quantitative PCR (RT–qPCR)

For RNA extraction, the tissue was snap-frozen in MP Lysing Matrix Tubes (containing beads for homogenization) and upon isolation supplemented with TRIzol and mechanically dissociated using a Ribolyser FastPrep-24 homogenizer (MP-Bio). RNA was isolated based on chloroform extraction. Isolated RNA was reverse-transcribed using Superscript III Reverse Transcriptase (Promega) to obtain cDNA for RT–qPCR. A QuantStudio 3 system (Applied Biosystems) was used for RT–qPCR with fluorescent SYBR Green detection. Thermocycling conditions used for RT–qPCR were as follows: hold (50°C, 2 min); hold (95°C, 10 min); 40 cycles of amplification (95°C, 0:15 min/X°C based on primer, 1 min); and final melting curve analysis: 95°C, 0:15 min; X°C based on primer, 95°C (dissociation), 0:15 min with a temperature increment of 0.1°C per second. Gene expression values were normalized to housekeeping genes *Gapdh* or *alpha-tubulin* (*Tuba1b*). Data were calculated as $2^{(-\Delta\Delta CT)}$, using universal mouse cDNA as a reference sample. Primers and primer sequences are listed in Table 2.

### Immunoblotting

For protein extraction, the tissue was snap-frozen in MP Lysing Matrix Tubes (containing beads for homogenization) and upon isolation supplemented with RIPA lysis buffer and mechanically dissociated using a Ribolyser FastPrep-24 homogenizer (MP-Bio). The protein concentration was measured using the bicinchoninic acid protein assay kit (#23225; Thermo Fisher Scientific), 40 µg of protein was mixed with reducing agent (NP009; Life Technologies) and lithium dodecyl sulfate sample buffer (NP007; Life Technologies), boiled, and loaded on gel to separate proteins. Proteins were subsequently transferred to a nitrocellulose membrane that was blocked with 5% BSA or 5% milk for 1 h before incubation with primary antibody (against PRDM16, FGF12, CACNA2D2, or β-TUBULIN; Table 2) overnight. The next day, the blot was washed and incubated with HRP-conjugated secondary antibody for 1 h at RT. Bound antibodies were detected using Pierce ECL Western Blotting Substrate or SuperSignal West Femto Maximum Sensitivity Substrate (Thermo Fisher Scientific) on a Bio-Rad ChemiDoc XRS+ molecular imager equipped with Image Lab software (Bio-Rad Laboratories). Bands were quantified using NIH ImageJ software, with β-TUBULIN staining as a loading control.

**Single-nucleus multiomics**

Both the isolation protocol of the single nuclei and the bio-informatics analysis of the data were based on Amoni et al (2023), with minor modifications related to our study.

### Isolation of nuclei

Seven-day-old pups were decapitated, and hearts were immediately removed and placed in PBS, all done on ice. The atria, part of the base, and the entire right ventricle were removed, and the remaining LV was cut into small pieces and snap-frozen in liquid $N_2$. Four samples per genotype were pooled to obtain ~80 mg of heart tissue per sample into gentleMACS M Tubes containing lysis buffer (5 mM $CaCl_2$, 3 mM MgAc, 2 mM EDTA, 0.5 mM EGTA, and 10 mM Tris–HCl, pH 8, in $H_2O$, supplemented before use with 1 mM DTT, 1 $\mu g$/ml actinomycin D, 0.05% Protease Inhibitor Cocktail, and 0.04 U/$\mu$l RNA inhibitors—RNase OUT). The first step of the isolation protocol was a mechanical dissociation using the gentleMACS dissociator. After mechanical dissociation, the tissue was exposed to lysis buffer supplemented with NP-40 (0.1%) and digitonin (0.01%) for lysis of the cell membrane. After a 15-min incubation on ice, the homogenate was passed through a 30-$\mu m$ filter and spun down. The pellet was dissolved in sucrose buffer (1 M sucrose, 3 mM MgAc, and 10 mM Tris–HCl, pH 8, in $H_2O$, supplemented with 1 mM DTT, 1 µg/ml actinomycin D, 0.05% Protease Inhibitor Cocktail, and 0.04 U/$\mu$l RNA inhibitors—RNase OUT). To isolate the nuclei from the cell debris, the nucleus-containing sucrose solution was pipetted gently on top of fresh sucrose buffer. After centrifugation, nuclei were pelleted followed by multiple rounds of resuspension in washing buffer (750 $\mu g$/ml UltraPure BSA and 0.04 U/$\mu$l RNA inhibitors—RNase OUT in PBS) before sorting. Nuclei were sorted on an ARIA3 FACS sorter based on 7-AAD staining. Sorted nuclei were counted on a LUNA cell counter, spun down, and resuspended in washing buffer to the desired concentration according to the 10X Genomics recommendations.

### Multiomics library preparation

After FACS, samples were submitted to the KU Leuven Genomics Core and processed using the 10X Genomics Next GEM Single Cell Multiome Assay for Transposase-Accessible Chromatin (ATAC) + Gene Expression Reagents kit to generate the gel beads-in-emulsion (GEMs) containing single nuclei. Next, joint single-nucleus RNA and single-nucleus ATAC libraries were generated according to the 10X Genomics protocol. Finally, libraries were sequenced at a depth of 30 K on Illumina NovaSeq 6000 by the KU Leuven Genomics Core.

### Multiomics data analysis

Raw data were processed by 10X Cell Ranger ARC to demultiplex raw base call (BCL) files generated by Illumina sequencer into FASTQ files. The FASTQ files were subsequently aligned to the mouse reference genome mm10, and count matrices were generated for both RNA and ATAC molecules. These unfiltered count matrices were first processed by CellBender to remove ambient RNA and scDblFinder to remove doublets (Table S2). These filtered matrices were finally further processed by R packages Seurat v4.0 and Signac v1.10. Per sample (Prdm16^WT and Prdm16^cKO), a SeuratObject was generated that included an RNA data slot for gene expression and an ATAC data slot for chromatin fragments. Each SeuratObject was quality-checked based on the number of unique molecular identifiers per cell, number of genes detected per cell, genes per unique molecular identifier, mitochondrial RNA, nucleosome signal, and TSS enrichment score. The exact thresholds used for both RNA data and ATAC data slots on both samples are summarized in Table S2. RNA data were processed, and normalization was performed followed by principal component (PC) analysis, neighbor identification (k = 20), and Uniform Manifold Approximation and Projection (UMAP). The number of PCs used for UMAP embeddings was calculated as the last PC where the difference between two subsequent PCs was less than 0.1% in variation, being 11 PCs for Prdm16^WT and 15 PCs for Prdm16^cKO. After UMAP, clustering was performed using the smart local moving algorithm (Waltman & van Eck, 2013) at a low resolution of each sample to identify the main cell populations. These clusters were used to recall the peaks per sample using MACS2 based on a combined set of peaks. After recalling the peaks, ATAC data were normalized using the term frequency–inverse document frequency (TF-IDF) followed by top feature selection and singular value decomposition on the TD-IDF matrix. This dimension reduction of ATAC data is known as late semantic indexing. Late semantic indexing components were used for graph-based clustering of the ATAC data as described for the RNA data.

### Multiomics data integration

Datasets were merged and integrated for RNA and ATAC assays based on anchor integration for each assay, separately. The batch effect of the RNA datasets was corrected by reciprocal PC analysis (RPCA). Finally, RNAseq and ATACseq data were integrated using weighted nearest neighbor (WNN) analysis following the recommended Seurat vignette for 10X Multiome datasets. Dimensional reduction was performed on the integrated SeuratObject, now containing both RNA and ATAC assays, using the FindMultiModalNeighbors function of Seurat. PCs used to find neighbors and for UMAP afterward were again calculated as described above. Clustering was performed using the smart local moving algorithm, initially at low resolution (0.09) to identify the main cell clusters, then at high resolution (0.5 for MCs, 0.4 for ECs and CMs, and 0.1 for FBs and ICs) to identify subclusters. Cluster marker genes were identified using FindAllMarkers of the Seurat package. Thresholds were set at $Log_2FC > 0.25$, min.diff.pct > 0.1, only.pos = TRUE, and Pval_adj > 0.05. Cluster identities were annotated manually based on literature (references included in the main text). For EC and MC clusters, populations of 77 and 26 cells, respectively, were identified as contaminated (because of the mixed expression of marker genes of different populations) and were therefore removed from these clusters. Populations were compared using Fisher's exact test with an FDR < 0.05 as a threshold for significance.

### Downstream analysis on CMs

DEG and DAR analyses were performed using the FindMarkers function on either the RNA or ATAC data slot, respectively. DEG analysis was performed using the Wilcoxon test with thresholds Pval_adj < 0.05. DAR analysis was performed using logistic regression with the total number of ATAC fragments as a latent variable, with thresholds Pval_adj < 0.05 and min.pct = 0.05. The R

package clusterProfiler was used for GO and on the DEG lists, using all genes detected as a list of background genes. Additional GO analysis was done in ToppGene using the default settings. Analysis for GO was performed with the Benjamini–Hochberg Pval_adj method, p_val < 0.01, and q_val < 0.05. GO and functional annotations of DARs were performed using rGREAT with default parameters, filtered for p_val < 0.05. Annotation of the DARs was performed using HOMER. Finally, peak-to-gene linkage analysis was calculated for all associated peaks and genes using the built-in function of Signac with default parameters.

### Motif enrichment analysis

DARs were scanned for enriched motifs using the built-in function of Signac (chromVAR) and using HOMER. PRDM16 binding motif was constructed based on the CIS-BP database (Weirauch et al, 2014), and manually added to the HOMER library. Next, DARs were scanned by HOMER to identify potential PRDM16 DNA binding places.

### GRN analysis

FigR was used to build GRNs based on the multiome input data. From the complete merged cardiomyocyte Seurat_object, FigR first determines peak–gene associations based on our paired single-cell accessibility (ATAC) and RNA count data. Per gene, significant peak–gene associations are identified as DORCs. Next, genes with associated regulatory regions were filtered for being differentially expressed and for being known as either atrial-specific or conduction-specific, using previously established gene signatures for each (Table S7) (Shekhar et al, 2018; Cao et al, 2023). FigR then identifies TF modulators linked to these regulatory regions and genes to construct a GRN displaying TF–gene relationships. Default parameters were used, with a transcription regulatory mean score of 0.5. Heatmaps and networks were constructed following the FigR manual.

### Code

Non-custom analysis software was created to perform the analysis. Images were analyzed using Fiji and QuPath built-in tools. Multiome sequencing data were analyzed using standard R packages (R v4.2.3), referred to throughout the article and in Table 2.

### Microscopy and imaging analysis

Fluorescence images of Fig 1 (Cy3, Alexa 568, TO-PRO-3) were taken using a ZEISS LSM 700 laser scanning confocal microscope and a Plan-Apochromat 63x/1.40 Oil DIC M27 objective (Fig 1B) or a ZEISS microscope equipped with an AxioCam 506 mono camera and EC Plan-Neofluar 10x/0.30 M27 objective to make mosaics and reconstruct full sagittal heart sections (Fig 1E). Brightfield/polarized light images of Sirius Red–stained sections in Fig 1D were taken using a ZEISS microscope equipped with an AxioCam HRc camera and an EC Plan-Neofluar 10x/0.30 M27 objective for mosaic images to construct full heart sections and an EC Plan-Neofluar 20x/0.40 M27 objective for representative insets. Fluorescence images of Fig 2 (Alexa 568; Hoechst) were taken using a ZEISS microscope equipped with an AxioCam 506 mono camera and an EC Plan-

Apochromat 20x/0.8 M27 objective (Fig 2D). Fluorescence images of Fig 4F (Alexa 488) were taken with a ZEISS microscope using an AxioCam MRc5 camera and EC Plan-Neofluar 2.5x/0.075 M27 objective, or EC Plan-Neofluar 5x/0.15 M27 objective for insets. Fluorescence images of Fig S1 (Alexa 488, Alexa 568, TO-PRO-3; Hoechst) were taken using a ZEISS LSM 700 laser scanning confocal microscope using a Plan-Apochromat 20x/0.8 M27 objective (Fig S1G and I) or a Plan-Apochromat 63x/1.40 Oil DIC M27 objective (Fig S1H and J), or a ZEISS microscope equipped with an AxioCam 506 mono camera with Plan-Apochromat 40x/0.95 Korr M27 objective to make mosaics and reconstruct a full sagittal heart section (inset represents 1 tile of the mosaic; Fig S1E and F). Brightfield images of Fig S1 were taken using a ZEISS Axio Observer microscope equipped with an AxioCam MRc5 camera and EC Plan-Neofluar 20x/0.50 M27 objective (Fig S1B and C) or an EC Plan-Neofluar 40x/0.75 M27 objective (Fig S1D). Fluorescence images of Fig S2 (Alexa 488, Alexa 568; Hoechst) were taken using a ZEISS LSM 700 laser scanning confocal microscope with an EC Plan-Neofluar 10x/0.30 M27 objective (Fig S2B) or a Plan-Apochromat 63x/1.40 Oil DIC M27 objective (Fig S2C and E), or a ZEISS microscope equipped with an AxioCam 506 mono camera and an EC Plan-Neofluar 10x/0.30 M27 objective to make mosaics and reconstruct the full brain section (Fig S2D) or an EC Plan-Apochromat 20x/0.8 M27 objective (Fig S2D, insets) to make mosaics of the area of interest or a ZEISS microscope with an AxioCam 506 mono camera and an EC Plan-Apochromat 40x/0.95 Korr M27 objective (Fig S2F). Fluorescence images of Fig S3 (Alexa 488) were taken using a ZEISS Axio Observer microscope equipped with an AxioCam MRc5 camera and EC Plan-Neofluar 2.5x/0.075 M27 objective to make mosaics and reconstruct the full heart section. Brightfield images of Sirius Red–stained sections in Fig S4C and G were taken using a ZEISS microscope with AxioCam HRc camera and an EC Plan-Neofluar 10x/0.30 M27 objective for mosaic images to construct full heart sections and an EC Plan-Neofluar 20x/0.40 M27 objective for representative insets. Brightfield images of Fig S7C were taken using a ZEISS microscope equipped with an AxioCam 506 color camera and an EC Plan-Neofluar 10x/0.30 M27 objective to make mosaics and reconstruct full sagittal heart section. Fluorescence images of Fig S7D (Alexa 568; Hoechst) were taken using a ZEISS microscope with an AxioCam 506 color camera and EC Plan-Apochromat 20x/0.4 M27 objective to make mosaics and reconstruct full sagittal heart section. Imaging software connected to the microscope and camera was AxioVision or ZEN Blue. All images were analyzed using Fiji or QuPath as indicated throughout the article.

### Statistics

Continuous data were presented as the mean ± SEM. Continuous datasets were analyzed using GraphPad Prism v9.4.1. When both groups were normally distributed (Shapiro–Wilk test), a parametric two-sided unpaired $t$ test was used. Where not all groups in an experiment were normally distributed or when normality could not be estimated by the Shapiro–Wilk test, a non-parametric Mann–Whitney test was used. In case of comparing multiple groups for echocardiography parameters, a two-way ANOVA with a Bonferroni post hoc test was performed. The numbers of mice used for each

group were indicated in the (supplementary) figure legends and supplementary tables. Data were considered significant when $P < 0.05$ (in case of multiple comparison, i.e., when analyzing the sequencing data, corrected by the Benjamini–Hochberg procedure; referred to as $P_{adjusted}$ or FDR). Statistical analysis of the single-cell multiome data was done in R (v4.2.3) using the aforementioned packages Seurat, Signac, clusterProfiler, and FigR. Populations were compared using Fisher's exact test with an FDR < 0.05 as threshold for significance.

### Online supplemental materials

Table S1 summarizes echocardiography parameters of P7, 8- and 16-wk-old mice, including statistics and sample sizes. Table S2 shows the number of nuclei per condition after different quality control steps. Table S3 lists marker genes for the cellular landscape of P7 hearts and the DEGs of $Prdm16^{cKO}$ versus $Prdm16^{WT}$ FBs. Table S4 represents DEGs of $Prdm16^{cKO}$ versus $Prdm16^{WT}$ CMs. Table S5 represents DARs of $Prdm16^{cKO}$ versus $Prdm16^{WT}$ CMs. Table S6 represents GO terms related to both DEGs and DARs. Table S7 represents full lists of DEGs overlapping with atrial, ventricular, conduction, or working CMs, as well as lists of unique DEGs for atrial or conduction CMs. Table S8 represents full lists of DARs overlapping with atrial, ventricular, conduction, or working CMs. Table S9 represents a full list of TF–gene regulation as shown on bar graphs and heatmaps of Fig 5D and E. Fig S1 shows PRDM16 expression in the developing and early postnatal heart, and efficiency and specificity of $Prdm16$ deletion in the heart at P7 and SMC coverage of coronary arteries at P7. Fig S2 shows mouse models used in this study, the activity pattern of the SM22α-Cre reporter, and the specificity of $Prdm16$ deletion in brains and lungs at P7. Fig S3 shows the analysis of the compact layer thickness based on ENDOMUCIN staining. Fig S4 shows the cardiac phenotype of $Prdm16^{cKO}$ mice and their $WT$ littermates at 8 and 16 wks of age. Fig S5 shows UMAPs, marker genes, and cell proportions of re-clustered main cell populations of the heart; and $Prdm16$ expression in the different subclusters. Fig S6 shows the ventricular-to-atrial and working-to-conduction shifts based on DEGs and DARs. Fig S7 shows the validation of the expression of some DEGs by RT–qPCR or at the protein level (by immunohistochemistry or immunoblotting). Fig S8 highlights the PF cluster including the expression of known PF markers. Fig S9 shows TF motif analysis on all up-regulated DARs and highlights overlapping 31 motifs between chromVAR and HOMER. In addition, network analysis of TF–gene regulation using FigR is visualized, highlighting the genes uniquely regulated by PRDM16. Fig S10 shows the expression of markers typical for trabecular or compact CMs.

## Data Availability

Sequencing reads and single-nucleus expression matrices have been deposited in NCBI's Gene Expression Omnibus (http://www.ncbi.nlm.nih.gov/geo/). The accession number for the snRNAseq and snATACseq data reported in this study is GSE255382. Any additional information required to reanalyze the data reported in this study is available upon request from the lead contact. All data were analyzed with standard programs and packages as indicated in Table 2. Non-custom analysis software was created to perform the analysis. Further information on and requests for resources and reagents should be directed to and will be fulfilled by the lead contact, Aernout Luttun (aernout.luttun@kuleuven.be).

## Supplementary Information

## Acknowledgements

We thank P Vandervoort, E Caluwé, and M Lox (Center for Molecular and Vascular Biology, KU Leuven, Leuven, Belgium) for technical assistance, and K Sipido, R Vennekens, E Jones, and A Zwijsen for insightful discussions. We acknowledge the KU Leuven Genomics Core for RNA and ATAC sequencing. This work was supported by internal KU Leuven funding (C14/19/095 to A Luttun, a Program Financing grant PF/10/014 to A Luttun, and C14/21/093 to HL Roderick); a European Research Council Grant (FP7-StG-IMAGINED203291) to A Luttun; three Fonds voor Wetenschappelijk Onderzoek (FWO) grants (G099521N, G060923N, and WOG001420N to A Luttun); three predoctoral FWO fellowships (1157318N to W Dheedene, 1S25817N to J Van Wauwe, and 11P9W24N to P Vrancaert); and one postdoctoral FWO fellowship (1271824N to H Kemps). The multiomics studies were funded by the support of the Belgian Heart Fund, in collaboration with the Belgian Society for Cardiology, and managed by the King Baudouin Foundation (2022-J1190990-225777).

### Author Contributions

J Van Wauwe: conceptualization, data curation, formal analysis, validation, investigation, visualization, methodology, and writing—original draft, review, and editing.

A Mahy: formal analysis, investigation, methodology, and writing—review and editing.

S Craps: formal analysis, investigation, methodology, and writing—review and editing.

S Ekhteraei-Tousi: data curation, investigation, methodology, and writing—review and editing.

P Vrancaert: formal analysis, investigation, methodology, and writing—review and editing.

H Kemps: formal analysis, investigation, methodology, and writing—review and editing.

W Dheedene: formal analysis, investigation, methodology, and writing—review and editing.

R Doñate Puertas: data curation, investigation, methodology, and writing—review and editing.

S Trenson: investigation, methodology, and writing—review and editing.

HL Roderick: resources, investigation, methodology, and writing—review and editing.

M Beerens: conceptualization, formal analysis, supervision, investigation, methodology, and writing—review and editing.

A Luttun: conceptualization, resources, data curation, formal analysis, supervision, funding acquisition, and writing—original draft, review, and editing.

**Conflict of Interest Statement**

The authors declare that they have no conflict of interest.

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
