## [Reviewer comments · Life Science Alliance]

Life Science Alliance

PRDM16 determines specification of ventricular cardiomyocytes by suppressing alternative cell fates

Jore Van Wauwe, Alexia Mahy, Sander Craps, Samaneh Ekhteraei-Tousi, Pieter Vrancaert, Hannelore Kemps, Wouter Dheedene, Rosa Doñate Puertas, Sander Trenson, H. Llewelyn Roderick, Manu Beerens, and Aernout Luttun

DOI: <https://doi.org/10.26508/lsa.202402719>

Corresponding author(s): Aernout Luttun, KU Leuven

Review Timeline:

Submission Date:	2024-03-14
Editorial Decision:	2024-05-01
Revision Received:	2024-08-31
Editorial Decision:	2024-09-04
Revision Received:	2024-09-06
Accepted:	2024-09-09

Transaction Report:

May 1, 2024

Re: Life Science Alliance manuscript #LSA-2024-02719-T

Prof. Aernout Luttun
KU Leuven
Department of Cardiovascular Sciences Center for Molecular and Vascular Biology, Endothelial Cell Biology Unit
KU Leuven, Campus Gasthuisberg Onderwijs & Navorsing 1, Herestraat 49 box 911
Leuven B-3000
Belgium

Dear Dr. Luttun,

Thank you for submitting your manuscript entitled "PRDM16 determines specification of ventricular cardiomyocytes by suppressing alternative cell fates" to Life Science Alliance. The manuscript was assessed by expert reviewers, whose comments are appended to this letter. We invite you to submit a revised manuscript addressing the Reviewer comments.

Thank you for this interesting contribution to Life Science Alliance. We are looking forward to receiving your revised manuscript.

Sincerely,

B. MANUSCRIPT ORGANIZATION AND FORMATTING:

Reviewer #1 (Comments to the Authors (Required)):

In the manuscript, Wauwe et al. found that PRDM16 favors ventricular working cardiomyocyte identity, by opposing the activity of master regulators of ventricular conduction and atrial fate. Myocardial loss of PRDM16 during development resulted in hyperplasia of the ventricular conduction system. Even though the authors tried to be clear and sharp in their demonstrations, several points remained unclear and should be addressed.

Major Points:

1. Expect the results of immunohistochemistry, the expression level of Rrdm16 in both mRNA and protein levels detected in heart and other organs from P7, 8w and 16w mice is required to better demonstrate the deficiency of Rrdm16 in Prdm16 cKO mice.
2. To verify Rrdm16 loss was also evident in coronary arterial SMCs but not in coronary arterial ED, the mRNA and protein expression level of Rrdm16 need to be detected in coronary arterial SMCs and ED isolated from coronary arteries of Prdm16 cKO mice.
3. The relative fold changes of target genes detected by RT-qPCR were suggested to use the $2^{(-\Delta\Delta Ct)}$ method instead of ΔCt . In Fig.1C, S2B and S2F, the mRNA level of Nppa and Nppb need to be re-calculated. In Fig.4F, the mRNA levels of Prdm16 and Ryr3 were suggested to be represented by $2^{(-\Delta\Delta Ct)}$ instead of ΔCt .
4. Is there a gender difference in response to progressive cardiomyopathy or premature death caused by cardiac specific knockout of Prdm16 in the offspring heart? The authors need to address this question with additional in vivo data.
5. Did authors test whether the early-onset cardiomyopathy in Prdm16 cKO mice can be functionally restored by overexpression of Prdm16? This piece of data is very essential.
6. What exactly are the markers of the subcluster with a VCS signature? How can we know the subcluster with a VCS signature is strikingly overrepresented or even up to 30-fold increase in Prdm16 cKO hearts in Fig.S3A?
7. Did authors detect the shift towards atrial and conduction fate in heart samples from Prdm16 cKO and Prdm16 WT mice? To illustrate that PRDM16 loss in CMs causes a shift towards atrial and conduction cell fate, the expression of ventricular genes (Kcne1, Myh7b), atrial genes (Fgf12, Myl4), ventricular working CM genes (Kcnd2, Pde3a;) and ventricular conduction genes (Cacna2d2, Ryr3) need to be detected as well in the heart samples from Prdm16 cKO and Prdm16 WT mice by RT-qPCR and western blot.

Minor points:

1. In FIGURE LEGENDS, "mean {plus minus} sem" should be replaced as mean {plus minus} SEM accordingly. Also, in Supplemental Material_Figures Page 3 Line 58, "mean {plus minus} sem" should be corrected as mean {plus minus} SEM.
2. For better comparison, the FigS1H(left) is suggested to be labeled with heart muscle E9.5 and Cre+.
3. In Page 3 Line74 of Fig.S2, the figure legend (n=1/15 for EF) is incorrect, n=13/15 for EF?
4. The manuscript should be edited for some typographical errors:
Page 27 Line 789, "Continuous data are..." should be were.
Page 27 Line 794, "group are..." should be were.

Reviewer #2 (Comments to the Authors (Required)):

Van Wauwe et al have investigated the effect of cardiomyocyte specific deletion of PRDM16 on ventricular specification in the mouse heart. In a carefully performed and well documented multiomics study the authors observe a shift towards a trabecular fate associated with cardiomyopathy and conduction defects implicating PRDM16 in promoting working cardiomyocyte identity and providing novel molecular insights into the mechanisms by which Prdm16 impacts ventricular fate. In order to reinforce the

authors' conclusions the following points should be considered.

1. Expression of PRDM16 appears to initiate between E9.5 and E10.5. Can the authors provide any further information using PRDM16 immunofluorescence? Or use the lacZ allele to evaluate later expression cross the ventricular wall?
2. The authors say that Prdm16 conditional mutant hearts show clear signs of perivascular and interstitial fibrosis compared with littermate controls. From the representative images presented in Fig. 1D, it is hard to appreciate interstitial fibrosis in mutant animals. Can the authors provide any histological quantification of this point? Is this considered to be an indirect effect of Prdm16 deletion? Please comment on whether transcriptional changes are observed in the increased fibroblast population.
3. Concerning the transcriptomic analysis, can the authors strengthen their conclusion that PRDM16 regulates cell fate choice between working and conductive or atrial CMs with additional validation on DEGs they identify between the two genotypes? Were proliferative markers restricted to cardiomyocytes? Please show the expression of Prdm16 itself in the different clusters in wildtype and conditional mutant hearts.
4. The authors should better define the nature of CMs within cluster 0 and cluster 1 from the re-clustered analysis showed in Fig 3A and B. It seems that only a small portion of cluster 1 express markers of VCS such as *Cacna2d2* and *Cntn2* but the authors claim that this cluster is expressing uniquely VCS genes.
5. The authors suggest that PRDM16 may compete with TBX5 for DNA binding. Can the authors provide any more insights to support this interesting hypothesis?
6. Endomucin staining in Figure 1E looks very broad. Please verify that this is the correct antibody used (and not a pan-endothelial cell marker).

Minor points:

6. The term SMC coating rate implies dynamic quantification. Can the authors use a better term?

Reviewer #3 (Comments to the Authors (Required)):

Title

PRDM16 determines specification of ventricular cardiomyocytes by suppressing alternative cell fates

Summary

The manuscript investigates the role of transcription factor PRDM16 in cellular fate decisions. In the past years, over the course of several publications by other authors rightfully cited in this manuscript, it has become more apparent that it is very likely that Prdm16 plays an important role in the specification of ventricular cardiomyocytes. This paper builds further on that knowledgebase. Their Sm22-alpha-Cre conditional knock-out of Prdm16 allows for the postnatal investigation of the knock-out and to my knowledge the authors present the first chromatin accessibility data on a Prdm16 knock-out. The manuscript is well written and the figures and data are well laid-out.

PRDM16 loss in CMs during development causes early-onset cardiomyopathy / Prdm16 deletion in CMs during development leads to premature death or progressive cardiomyopathy (Figure 1 / Table 1)

The study demonstrates that PRDM16 loss in cardiomyocytes during development results in early-onset cardiomyopathy. This is supported by the measurements of ejection fraction, hypertrophy, and the ECG. Furthermore, the data show that this Prdm16 cKO can lead to premature death, but is not 100% lethal after birth.

Figure 1B shows a quantification of hypertrophy of cells as measured by their area, could you show a representative image?

What could be the mechanism of hypertrophy caused by Prdm16 cKO, are atrial and / or conduction cells larger than ventricular cells?

Figure 1C uses Nppa / Nppb expression as a proxy for cardiac stress, however these are also markers for unstressed atria.

Since later in the paper you dive into this atrial-shift further, why would these markers be a measure of cardiac stress?

Figure 1D shows an increase in fibrosis in the cKO. Why would the hearts develop fibrosis without any injury or excessive stress?

Figure 1E shows immunofluorescence stainings of endomucin. Due to the high intensity detected outside of the endocardium, is this marker the best way to determine the trabeculae/lumen interface? The lines drawn do not clearly correspond to the detected. The endomucin signal seems stronger in the cKO, are these imaged at the same settings? Could the slight increase in endothelial cells explain this difference? Furthermore, the way the compact layer thickness is quantified is unclear.

PRDM16 loss in CMs during development perturbs the cardiac cellular landscape (Figure 2)

The study performed multiome-sequencing on the cKO and the WT at P7 to investigate the transcriptome and chromatin accessibility. The manuscript states that PRDM16 loss in cardiomyocytes during development perturbs the cardiac cellular landscape.

Figure 1A shows the UMAP of the integrated datasets, however perhaps a UMAP with the separate replicates could be added to confirm the clustering is not due to inter-replicate variability.

Figure 1C shows that there is a shift in the fractions of different cell types present in the cKO when compared to the fractions in the WT. However, it is best to be careful with quantifications like these because the isolation has a strong impact on which cell types are preferentially isolated. The same protocol on the WT and the cKO could lead to different fractions purely due to the isolations. Could an experiment be added that quantifies cell types using immunofluorescent stainings?

PRDM16 loss in CMs triggers changes in gene expression and chromatin accessibility related to hypertrophy, metabolism, conduction, and TGF β signaling (Figure 3)

The study provides data showing that PRDM16 loss in cardiomyocytes triggers changes in gene expression and chromatin accessibility. The data show that the 'proliferative' cardiomyocytes overlap between the WT and the cKO, but the cluster of 'mature' cardiomyocytes is greatly diminished in the cKO. These data support the role of PRDM16 in regulating key pathways involved in cardiac function and cellular identity.

The m+pCM nomenclature was taken from the clusters found in figure 2, however since this zoomed-in clustering still results in a mature and a proliferative cluster, this nomenclature is unclear.

All clusters of cardiomyocytes of the WT and the cKO are pooled to perform DEG analysis. This would be expected to dilute differential signals by adding in the cells with a similar identity (cluster 2 and part of cluster 1). Wouldn't it be interesting to compare cluster 0 and cluster 1 (perhaps with the WT Purkinje cells excluded). Further, you mention hyperplasia of the ventricular conduction system, could you perform a direct comparison between the cluster 1 cells from the cKO and the very small portion of cluster 1 WT cells that are supposed to represent the VCS cells in a healthy mouse. This way you could perhaps identify in what way these cells are and are not similar to the Purkinje fibres.

Rather than using upregulated and downregulated for the volcano plots, use higher and lower expressed. Regulation implies an active process which, while it's possible that this is the case here, is not shown using two static captures of separate transcriptomes. Clearly communicate that each volcano plot is the same plot, and do not use different a x-axis for one of them. The number of DEGs found to be lower expressed in the cKO is lower, therefore it is difficult to assess whether the fraction of genes for the gene-sets tested is actually that different. Could you add a functional annotation of KEGG pathways or a different measure, and perform the statistical enrichment.

PRDM16 loss in CMs causes a shift towards atrial and conduction cell fate (Figure 4)

The study asserts that PRDM16 loss in cardiomyocytes causes a shift towards atrial and conduction cell fate. This is supported by the identification of differentially expressed genes associated with atrial-specific and VCS-specific genes in PRDM16-deficient cardiomyocytes, indicating a deviation from the normal ventricular working cardiomyocyte identity.

Figure 4A shows the expression pattern of certain genes classified as atrial to be expressed in the small population of Purkinje cells of the WT (e.g. Myl4, Fgf12). Could more convincing data be presented to imply that this is a shift to both atrial and VCS, rather than just VCS. Seemingly there is overlap between atrial and VCS specific properties. Furthermore, could also be elaborated on the ways these cells are not similar to healthy WT Purkinje cells and atrial cells (although that would understandably be more complicated due to the lack of atrial cells in the dataset)?

Figure 4H shows two images of a Cntn2 staining, which is nicely restricted to the Purkinje. However, do these sections represent a similar depth cut through the heart. It could be contractile state, but it looks like the cavity size is small due to being a fairly shallow cut in the WT. Furthermore, the quantification looks much too high, the cKO image does not show 10% of the total ventricular area to be Cntn2 positive. It does look like an increase, but roughly 10% sounds very unlikely. Did the measurement take the folds with strong background into account?

Finally, in the discussion (line 429) the following is written "in CMs we did not see elaborate transcriptomic signs of a switch towards trabecular fate upon PRDM16 loss". Could a figure be added that shows this observation?

PRDM16 suppresses the activity of master regulators of atrial and conduction fate (Figure 5)

The research findings suggest that PRDM16 suppresses the activity of master regulators of atrial and conduction fate. This is supported by the analysis revealing that PRDM16 mediates its repressive effects on atrial and conduction marker genes through opposition of master regulator transcription factors, particularly TBX5, highlighting the regulatory role of PRDM16 in cardiomyocyte fate determination.

Line 314 states that "PRDM16 puts a double brake on their activity by lowering both their expression and accessibility to DNA" however, doesn't lowered accessibility necessarily decrease the expression anyway, how can these two mechanisms be separated?

For the heatmaps in figure 5D and E it is unclear what exactly these represent. It would benefit from a more comprehensive explanation as to what the axes represent and which variable is presented by the colour-scale.

PRDM16 orchestrates CM fate decision by acting on promoters and distant enhancers (Figure 6)

The study provides evidence that PRDM16 orchestrates cardiomyocyte fate decisions by acting on promoters and distant enhancers. This is supported by combining their ATAC data with previously published ChIP-seq data to identify direct and indirect PRDM16 binding sites in the promoter regions of atrial and conduction genes.

Figure 6A shows a decision tree for identifying mechanisms of action for PRDM16. Why was the ChIP-seq data not also checked for signal in the identified distal enhancer elements?

Overall Impression

The manuscript significantly advances our understanding of PRDM16's role in cardiomyocyte specification and fate determination. However, certain points require further clarification and validation, particularly regarding mechanistic insights and the specificity of observed effects.

Rebuttal manuscript N°LSA-2024-02719-T entitled 'PRDM16 determines specification of ventricular cardiomyocytes by suppressing alternative cell fates' by Van Wauwe *et al.*

Changes made to the manuscript not directly related to the Reviewers' concerns:

- Due to the revision process requiring additional experimental work, the author list has changed with altered positions for two authors (A. Mahy and P. Vrancaert) and the addition of one new author (H. Kemps). The author contribution section has been adapted accordingly (line 1132).
- Keywords and a summary blurb have been added.
- The running title has been shortened to meet formatting requirements.
- A mistake in **Table S2** has been corrected.
- For consistency, 'Purkinje cells' has been systematically replaced by 'Purkinje fibers' (PF).
- The acknowledgements section has been updated (lines 1123-1126).

NOTES:

- **Please note that the line numbers to which we refer in this rebuttal correspond to those in the pdf file of the version with track changes.**
- Figures included in this rebuttal for the Reviewers' interest only are labeled 'R'.
- Figure numbering has altered due to rearrangement of and addition of new (supplementary) figures in the revised manuscript and supplement.
- Actions taken to modify main text/supplement in response to the Reviewers' questions are underlined in this rebuttal.
- The sections 'Microscopy and imaging analysis' (lines 1012-1054) and 'Online supplemental materials' (lines 1071-1096) have been updated according to the addition of new data.

Point-by-point answer to the Reviewers' concerns:

A. Reviewer 1

1. Expect the results of immunohistochemistry, the expression level of *Prdm16* in both mRNA and protein levels detected in heart and other organs from P7, 8w and 16w mice is required to better demonstrate the deficiency of *Prdm16* in *Prdm16* cKO mice.

Response: Given the variable results obtained with different Cre-driver lines reported by previous studies (cited in the introduction of the original and revised manuscript (Cibi DM et al, 2020, Kramer RJ et al, 2023, Nam JM et al, 2020, Wu T et al, 2022)), we agree with the Reviewer that it is very important to thoroughly demonstrate the efficiency (and specificity) of *Prdm16* deletion in our conditional knockout (KO) model. While efficient and specific *Prdm16* deletion in PCM1⁺ cardiomyocytes in the P7 hearts was shown by immunohistochemistry (IHC) in **Figure S1F,G,J,K** of the original manuscript, and the corresponding quantification of deletion efficiency was mentioned on line 142 of the main manuscript, we have now provided additional evidence for efficient *Prdm16* deletion in P7 hearts by using two additional detection methods (*i.e.* RT-qPCR and immunoblotting on whole left ventricle tissue lysates). Furthermore, to further confirm the specificity of the Cre-driven approach, we also looked by IHC into other P7 organs, where deletion was restricted to arterial SMCs and thus not apparent in other cells known to express PRDM16 (*e.g.* lung bronchial epithelial cells (Fei LR et al, 2019) and various cell types in the subventricular zone and the choroid plexus in the brain (Shimada IS et al, 2017, Strassman A et al, 2017) and (arterial) endothelial cells (Craps S et al, 2021)). Since our manuscript focusses mostly on the P7 time point, we have compiled all P7 *Prdm16* deletion

data for the heart in revised Figure S1, and all P7 *Prdm16* deletion data for brain and lung in a novel Figure S2. In addition, for the Reviewer's interest, we also provide here an account of our additional stainings in heart, brain and lungs at 8 weeks (w) and 16w, confirming the specificity of the deletion (Figure R1). Our additional stainings on the 8w or 16w hearts also reveal that PRDM16 is barely detectable in *WT* cardiomyocyte (CM) nuclei (in line with findings by Wu *et al.* (Wu T et al, 2022)), such that the expression level is the same as in the *cKO* CM nuclei (as opposed to P7, where expression was still clearly apparent in *WT* CMs and efficiently erased in *cKO* CMs). We have expanded the results section (lines 153, 155-157) and methods section (lines 723, 726-727) with the new experiments.

Figure R1. Deletion of *Prdm16* in vascular smooth muscle cells of the heart, brain and lung, but not in arterial endothelial cells and various other cell types. (A-F) Cross-sections stained for PRDM16 (green) and smooth muscle cell α -actin (α SMA; red) in the brain (A,B,D,E) or the lung (C,F) at the age of 8 weeks (w; A-C) or 16 w (D-F). (G-I) Cross-sections stained for PRDM16 (green) and/or α SMA (red) in the heart at the age of 8w (G,H) or 16w (I). Note the absence of a clear nuclear expression pattern for PRDM16 at 8w and 16w in the myocardium (as opposed to P7, shown in the inset). Nuclei are stained with Hoechst in G-I (left panels). Dotted white lines in A delineate the ventricle border, those in B,G delineate the intima/media border. Abbreviations: Br: bronchiole; art: artery; Cp; choroid plexus; SVZ: subventricular zone. Scale bars: 10 μ m (G), 20 μ m (A-F,H,I).

- To verify *Prdm16* loss was also evident in coronary arterial SMCs but not in coronary arterial ED, the mRNA and protein expression level of *Rrdm16* need to be detected in coronary arterial SMCs and ED isolated from coronary arteries of *Prdm16* cKO mice.

Response: We concur that for an additional confirmation of the specificity and efficiency of *Prdm16* deletion in arterial smooth muscle cells and not endothelial cells in P7 hearts, complementing the IHC data from **Figure S1** and novel **Figure S2** with RT-qPCR or immunoblotting on sorted coronary smooth muscle cells and endothelial cells would be reassuring. Nevertheless, given the low number of vascular cells available from a P7 heart (< 2% of total cells for arterial endothelial cells and < 1% of total cells for (arterial) smooth muscle cells; *cf.* **Figure S3** of the original manuscript), it is technically (too) challenging to sort them with sufficient yield to isolate RNA and protein of sufficient quality and quantity to detect and quantify *Prdm16*/PRDM16 by RT-qPCR or immunoblotting, respectively. Additionally, in light of your Question 1 regarding the efficiency of *Prdm16* deletion in cardiomyocytes, we performed a pilot experiment to sort cardiomyocyte nuclei based on PCM1⁺ immunostaining, however we experienced low yield (~70.000 PCM1⁺ nuclei for the ventricles of 1 heart) after sorting which made it technically challenging to measure gene expression or protein levels (see **Figure R2** included for the Reviewer's interest). Given the failure of detecting RNA expression from the most abundant cell type in the heart, we anticipate that obtaining sufficient RNA and protein from much smaller cell populations like coronary smooth muscle cells or endothelial cells would be even more challenging. To provide additional proof of consistent, specific and efficient deletion of *Prdm16* from arterial smooth muscle cells and not endothelial cells in different organs including the heart, we now provide more IHC data in the brain and the lung at different ages, which we included in novel **Figure S2** in the revised manuscript and in **Figure R1** above.

Figure R2. RT-qPCR amplification plots for *Prdm16* (left panels) and *Tuba1b* expression (right panels) revealing that while *Prdm16* and housekeeping gene *Tuba1b* could be detected in the reference sample (i.e., universal mouse cDNA; bottom panels), there was no signal for the P7 wild-type (WT) heart sample (top panels), suggesting there was not enough cDNA of good quality for RT-qPCR.

3. The relative fold changes of target genes detected by RT-qPCR were suggested to use the $2^{(-\Delta\Delta Ct)}$ method instead of ΔCt . In Fig.1C, S2B and S2F, the mRNA level of Nppa and Nppb need to be re-calculated. In Fig.4F, the mRNA levels of Prdm16 and Ryr3 were suggested to be represented by $2^{(-\Delta\Delta Ct)}$ instead of ΔCt .

Response: We agree with the Reviewer's comment on using $2^{(-\Delta\Delta Ct)}$, which is a better representation of the data. We have adapted all RT-qPCR-related panels from the original manuscript (**Figure 1C, S2B,F**, the latter being **S4B,F** in the revised manuscript) and have also used this method for the additional RT-qPCR data newly provided in the revised manuscript as an answer to Reviewers' questions (**Figure S1L, S7A, S10A,B**). We adapted the methods section (lines 890-891) accordingly.

4. Is there a gender difference in response to progressive cardiomyopathy or premature death caused by cardiac specific knockout of Prdm16 in the offspring heart? The authors need to address this question with additional in vivo data.

Response: Given recent publications reporting on sexual dimorphism (*i.e.* a higher prevalence and severity in females) related to the cardiac/vascular phenotype in cardiomyopathy patients with *PRDM16* mutations (Kramer RJ et al, 2023), and some more recent evidence from mouse models with cardiomyocyte-specific deletion of *Prdm16* reporting on a more severe phenotype in females (Kramer RJ et al, 2023, Kuhnisch J et al, 2024, Nam JM et al, 2020), we appreciate and acknowledge the Reviewer's question. In **Table 1** of the original manuscript, we already reported that there was no sex difference in terms of premature death, which is in agreement with Wu *et al.* (Wu T et al, 2022) who also did not see a sex difference in mortality before P7. To look into potential sex differences in cardiomyopathy parameters in the surviving mice, we now have looked at ejection fraction (EF; a parameter for systolic dysfunction) in the two sexes and did not find a significant difference in effect size between males and females, both at 8 and 16 weeks of age. We now show the split data in revised **Table S1**. The results section (line 193) and the section on statistics (lines 1060-1062) were updated accordingly. One potential explanation is the overruling of sex differences when *Prdm16* deletion leads to a severely dramatic cardiac phenotype. In the mouse models of Wu *et al.*, 100% lethality was observed by P7. In addition, although we only observed 60% lethality by weaning age in our mouse model, we also did not find sex dimorphism suggesting that severe phenotypes leading to perinatal or postnatal lethality blunt out the sex differences seen in *Prdm16* deficient models with a more mild (non-lethal) phenotype (Kramer RJ et al, 2023, Kuhnisch J et al, 2024, Nam JM et al, 2020). We have added a brief notion in the revised discussion (lines 525-532).

5. Did authors test whether the early-onset cardiomyopathy in Prdm16 cKO mice can be functionally restored by overexpression of Prdm16? This piece of data is very essential.

Response: We agree with the Reviewer that this would be a very interesting and relevant experiment to deliver additional proof of the causal relationship between the phenotype observed and the role of PRDM16 in cardiomyocytes and even to address the therapeutic potential of PRDM16. However, to achieve this would require a significant amount of time and resources. Indeed, in order to rescue the phenotype, a cardiomyocyte-specific PRDM16 overexpression approach would be required. Currently, no such approach (*e.g.* mice with the *SM22 α -Cre* driving the expression of a cassette with a floxed stop codon preceding a full-length *Prdm16* gene in the

Rosa locus, that then should be intercrossed with our *SM22 α -Cre-Prdm16^{fl/fl}* knockout model; or an adeno-associated virus (AAV) or lentiviral vector encoding full-length PRDM16 that specifically and efficiently targets cardiomyocytes from a very early stage - *i.e.* E9.5 - of development onwards) is currently available and validated. We hope the Reviewer agrees that this could well be the subject of a follow-up paper on the current manuscript.

6. What exactly are the markers of the subcluster with a VCS signature? How can we know the subcluster with a VCS signature is strikingly overrepresented or even up to 30-fold increase in *Prdm16* cKO hearts in Fig.S3A?

Response: We believe the Reviewer is referring to sub-cluster 1 in **Figure 3** and accompanying **Figure S3** of the original manuscript. First, we already provided the full marker gene panel for this sub-cluster in **Table S3** of the original manuscript. Second, we would like to apologize for the confusion we created by inappropriately labeling this sub-cluster as a cluster with a ventricular conduction system (VCS) signature. Indeed, even though this sub-cluster has a gene signature that is highly enriched for genes known to be expressed in the VCS, the same sub-cluster also shows enrichment for genes known to be associated with atrial cardiomyocytes.

Furthermore, many of the genes within the atrial signature **overlap** with genes in the VCS signature, as we point out in our answer to Question 10 of Reviewer 3. Given the mixed nature of the expression shifts in this sub-cluster, we should not merely label it as a VCS cluster, neither should we claim that there is a 30-fold expansion of the VCS cluster in the *Prdm16* cKO mice. We hence systematically revised the manuscript to avoid such a misleading statement (lines 244-247, 269, 277, 278; **Figure S5A** now shows a more appropriate set of marker genes comprising both atrial and conduction markers). What we can rightfully claim is that based on the CONTACTIN-2 immunohistochemistry analysis (which we revisited as an answer to Question 11 of Reviewer 3 and now report in revised **Figure 4F**), we do see a 5.4-fold expansion of the Purkinje fiber network, which is the most distal part of the VCS. Therefore, in the revised manuscript, we reworded 'VCS hyperplasia' as '(distal) VCS hyperplasia' (lines 53, 131, 382, 513, 549, 852; Figure 7 and graphical abstract).

7. Did authors detect the shift towards atrial and conduction fate in heart samples from *Prdm16* cKO and *Prdm16* WT mice? To illustrate that PRDM16 loss in CMs causes a shift towards atrial and conduction cell fate, the expression of ventricular genes (*Kcne1*, *Myh7b*), atrial genes (*Fgf12*, *Myl4*), ventricular working CM genes (*Kcnd2*, *Pde3a*;) and ventricular conduction genes (*Cacna2d2*, *Ryr3*) need to be detected as well in the heart samples from *Prdm16* cKO and *Prdm16* WT mice by RT-qPCR and western blot.

Response: As mentioned above, we indeed did see a double fate shift (*i.e.* towards atrial and conduction) in *Prdm16* cKO versus *Prdm16* WT P7 cardiomyocytes, as reported in **Figure 4** of the original manuscript. For some markers, we already complemented our single-nuclei expression data with IHC (*i.e.* for *MYL4*) or RT-qPCR data (*i.e.* for *Ryr3*), as reported in **Figure 4E,F** of the original manuscript. We nevertheless concur with the Reviewer that to make a more compelling case, we should perform additional validation studies by RT-qPCR, IHC and immunoblotting. We have added an additional supplementary **Figure S7** in which we have now merged all (existing and newly generated) validation results. In addition to the IHC data on *MYL4* and the RT-qPCR data on *Ryr3*, we now added RT-qPCR data on most of the gene

panel for which we showed the UMAP in original **Figure 4A,B** and on an additional gene (*i.e.*, *Ank2*). Furthermore, we also validated expression differences for some of them at the protein level by either immunoblotting (CACNA2D2, FGF12) or IHC (FHL2). We have expanded the methods section with the new experiments (lines 862-865, 870, 873-878, 893-908), updated **Table 2** and expanded the results section (lines 368-370).

8. Minor comments

- a. In FIGURE LEGENDS, "mean plus minus sem" should be replaced as mean plus minus SEM accordingly. Also, in Supplemental Material_Figures Page 3 Line 58, "mean plus minus sem" should be corrected as mean plus minus SEM.
- b. For better comparison, the FigS1H(left) is suggested to be labeled with heart muscle E9.5 and Cre+.
- c. In Page 3 Line74 of Fig.S2, the figure legend (n=1/15 for EF) is incorrect, n=13/15 for EF?
- d. The manuscript should be edited for some typographical errors:
 - i. Page 27 Line 789, "Continuous data are..." should be were.
 - ii. Page 27 Line 794, "group are..." should be were.

Response: We thank the Reviewer for pointing out these errors. We have corrected all of them in the revised manuscript. Note that for item b, the change has been in **Figure S2B** (as the figure composition has changed upon revision).

B. Reviewer 2

1. Expression of PRDM16 appears to initiate between E9.5 and E10.5. Can the authors provide any further information using PRDM16 immunofluorescence? Or use the lacZ allele to evaluate later expression cross the ventricular wall?

Response: We thank the Reviewer for the suggestion to complement our current expression pattern data for PRDM16 at E9.5 and E10.5 with expression analysis at later time points during development to better document its (asymmetric) expression pattern across the ventricular wall, as reported earlier by Wu *et al.* (Wu T et al, 2022). We therefore have set up timed matings to generate E14.5 and E17.5 pups and evaluate PRDM16 expression by combined immunofluorescence staining with ENDOMUCIN (to delineate the trabecular zone) on frozen cross-sections of the ventricular wall at both (later) time points (which were not yet reported by Wu *et al.*). We confirm that expression of PRDM16 in the ventricular wall is enriched in the compact myocardium compared to the trabecular myocardium, both at E14.5 and E17.5 (although less evident at E17.5 since the trabeculae start to disappear in *WT* mice). We have added these extra expression data to revised **Figure S1E,F**. We have expanded the methods section with the new experiments (lines 720, 726-727, 780-783) and updated the results section (lines 140, 141).

2. The authors say that Prdm16 conditional mutant hearts show clear signs of perivascular and interstitial fibrosis compared with littermate controls. From the representative images presented in Fig. 1D, it is hard to appreciate interstitial fibrosis in mutant animals. Can the authors provide any histological quantification of this point? Is this considered to be an indirect effect of Prdm16 deletion? Please comment on whether transcriptional changes are observed in the increased fibroblast population.

Response: We apologize for the suboptimal quality of the Figure panel showing more interstitial fibrosis in the mutant hearts at P7. We therefore have provided alternative images in revised **Figure**

1D based on Sirius Red stainings on additional P7 hearts. We did provide a quantitative analysis of the Sirius Red stainings on the right side of original Figure 1D. In addition, for the alternative interstitial fibrosis images, we now also provide in revised Figure 1D the corresponding area imaged by polarized light microscopy in order to even better display the presence of interstitial fibrosis in the mutant hearts. We have adapted the methods text (lines 832-833) and the figure legend (line 1162) accordingly. We concur with the Reviewer that this fibrosis is most plausibly an indirect effect of the cardiac dysfunction induced by *Prdm16* deletion, since it is unlikely that cardiomyocytes would be the producers of the collagen. Instead, the increase in the fibroblast population in the mutant hearts (as reported in Figure 2C of the original manuscript) likely is responsible for the increased fibrosis. When comparing the expression pattern of the fibroblast population between genotypes, then we clearly notice an increased expression of genes encoding extracellular matrix components, including collagens, and smooth muscle cell markers, together corresponding to a more myofibroblast-like signature. The differential expression analysis of the fibroblast population has been included in Table S3 of the revised manuscript and the results section has been expanded accordingly (lines 233-234).

3. Concerning the transcriptomic analysis, can the authors strengthen their conclusion that PRDM16 regulates cell fate choice between working and conductive or atrial CMs with additional validation on DEGs they identify between the two genotypes? Were proliferative markers restricted to cardiomyocytes? Please show the expression of *Prdm16* itself in the different clusters in wildtype and conditional mutant hearts.

Response: We thank the Reviewer for the suggestion to validate more atrial and conduction markers, which was a request commonly made by Reviewer 1 (Question 7). Therefore, we performed additional **validation** studies by RT-qPCR, IHC and immunoblotting. We have added an additional supplementary Figure S7 in which we have now compiled all (existing and newly generated) validation results. In addition to the IHC data on MYL4 and the RT-qPCR data on *Ryr3*, we now added RT-qPCR data on most of the gene panel for which we showed the UMAP in original Figure 4A,B and on an additional gene (*i.e.*, *Ank2*). Furthermore, we also validated expression differences for some of them at the protein level by either immunoblotting (CACNA2D2, FGF12) or IHC (FHL2). We have expanded the methods section with the new experiments (lines 862-865, 870, 873-878, 893-908), updated Table 2 and expanded the results section (lines 368-370).

As for **proliferation**, our omics data suggest that proliferation was not limited to cardiomyocytes; there were also subclusters expressing proliferation markers in other cell types, *i.e.* fibroblasts, endothelial cells and mural cells, as was shown in Figure S3 of the original manuscript.

We appreciate the question to show the **expression** of *Prdm16* in the different *WT* as well as *cKO* clusters (as another confirmation of successful deletion in cardiomyocytes and smooth muscle cells but not arterial endothelial cells), however, this leads to confusing results. The *cKO* of *Prdm16* was achieved by deletion of *exon 9*, as shown in Figure S1B of the original manuscript (Figure S2A in the revised manuscript). Hence, all other exons and introns of the *Prdm16* gene are still mapped, which seemingly results in *Prdm16* “expression” if shown on a violin plot. To make this clearer, for the Reviewer’s interest, we have performed extra bioinformatic analysis to get Bam files of the cardiomyocyte *WT* and *cKO* cluster, and visualized these in Integrative Genome Viewer. This clearly shows reduced reads for *Prdm16* *cKO* mice, and the absence of mapped reads in *exon 9* of *Prdm16* *cKO* nuclei (Figure R3). This type of

analysis is only clear when the clusters contain a sufficient number of nuclei (reads), which is why we could only do this for the cardiomyocyte cluster and not the arterial endothelial or smooth muscle cell cluster. We hope the Reviewer also appreciates our revised **Figure S1** and novel **Figure S2**, where we added additional proof of *Prdm16* deletion in our mouse model, as requested by Reviewer 1 (Question 1).

Figure R3. Integrative Genome Viewer plots of the region containing *Prdm16* exon 9 based on the ATACseq data (top panel) or RNAseq data (bottom panel).

- The authors should better define the nature of CMs within cluster 0 and cluster 1 from the re-clustered analysis showed in Fig 3A and B. It seems that only a small portion of cluster 1 express markers of VCS such as *Cacna2d2* and *Cntn2* but the authors claim that this cluster is expressing uniquely VCS genes.

Response: We apologize for the confusion we caused by inappropriately labeling sub-cluster 1 as a ‘VCS cluster’. Indeed, even though this sub-cluster has a gene signature that is highly enriched for genes known to be expressed in the VCS, the same sub-cluster also shows enrichment for genes known to be associated with atrial cardiomyocytes. Furthermore, many of the genes within the atrial signature overlap with genes in the VCS signature, as we point out in our answer to Question 10 of Reviewer 3. Given the mixed nature of the expression shifts in this sub-cluster, we should not merely label it as a VCS cluster, but rather as a ‘mixed’ sub-cluster. We hence systematically revised the manuscript to avoid such a misleading statement (lines 244-247, 269, 277, 278; **Figure S5A** now shows a more appropriate set of marker genes comprising both atrial and conduction markers). The reason why only a small portion of sub-cluster 1 shows expression of *Cntn2* and *Cacna2d2* is related to the fact that these markers are more prominent in Purkinje fibers which only represent a (small) subset of cells within the entire VCS. As we show in **Figure 4E** and **Figure S8** of the revised manuscript, cells with a Purkinje fiber signature indeed represented only a minor population within sub-cluster 1. To better emphasize this, we

have reorganized Figure 4 and Figure S8 and adapted the results text accordingly (lines 372-376, 379).

5. The authors suggest that PRDM16 may compete with TBX5 for DNA binding. Can the authors provide any more insights to support this interesting hypothesis?

Response: We thank the Reviewer for his/her appraisal of our hypothesis which was based on the observation that putative PRDM16 DNA binding sites were often colocalizing with sites for TBX5 binding to DNA. To deliver proof that this competition scenario is indeed occurring, this would require to set up and optimize competition binding assays, which would require a significant amount of time and resources. Although we consider this as an interesting path for further research, in the current absence of experimental proof for this hypothesis we propose to omit it from the main text and have reworded the text accordingly in the results section (line 493) and the discussion (lines 648-654 and deletion at line 654).

6. Endomucin staining in Figure 1E looks very broad. Please verify that this is the correct antibody used (and not a pan-endothelial cell marker).

Response: We apologize for the confusion. We are confident that we used the correct antibody, as the staining pattern we obtained is as expected and corresponding to what others have shown using ENDOMUCIN staining as a guidance to line the endocardium and visualize the trabeculae. Indeed, although ENDOMUCIN staining has been frequently used for this purpose, (Feng Q et al, 2010, Liu H et al, 2023, Lu P et al, 2023, Sun B et al, 2023, Wu T et al, 2022), it is not a marker specific for endocardial endothelium, but rather a more general endothelial marker. In support of this, *Emcn* was expressed in all endothelial sub-clusters (as shown here for the Reviewer's interest in Figure R4).

Figure R4. Uniform Manifold Approximation and Projection (UMAP) diagrams revealing expression of *Emcn* across all endothelial sub-clusters (right panel) while *Npr3* was restricted to the endocardial sub-cluster 2 (middle panel).

Nevertheless, we acknowledge that it might be more appropriate to use a more specific endocardial stain to measure trabeculation and the thickness of the compact layer. Therefore, we repeated the analysis using a more specific and restricted marker, *i.e.* NPR3 (encoded by one of the signature genes used for the endocardial sub-cluster 2 in Figure S3 or the original manuscript and shown here on an UMAP plot in Figure R4 for the Reviewer's interest), and confirmed/validated the data we obtained earlier based on ENDOMUCIN staining. Although we are still confident in our results based on ENDOMUCIN, we replaced these with the newly obtained analysis based on NPR3 staining in revised main Figure 1E. As additional validation, we have now moved the ENDOMUCIN-based analysis to the supplement (Figure S3) of the

revised manuscript. We have expanded the methods section with information on the NPR3 staining (lines 836 and 840-847), provided more details in the methods section of the revised manuscript (lines 850-851) and adapted the figure legend (line 1165, 1168).

7. Minor comments

- a. The term SMC coating rate implies dynamic quantification. Can the authors use a better term?

Response: We thank the Reviewer for pointing this out. In the revised manuscript and corresponding figure panel **Figure S1N**, we have now replaced the term 'coating (rate)' by 'coverage' (line 184 in the main text and line 29 in the legend to **Figure S1**).

C. Reviewer 3

1. Figure 1B shows a quantification of hypertrophy of cells as measured by their area, could you show a representative image? What could be the mechanism of hypertrophy caused by *Prdm16* cKO, are atrial and / or conduction cells larger than ventricular cells?

Response: We have now added representative images of cardiomyocyte hypertrophy (LAMININ staining) of *WT* and *cKO P7* hearts to revised **Figure 1B** and adapted the figure legend (lines 1157-1158). This hypertrophy could either be a pathophysiological response due to the cardiac dysfunction present as a consequence of poor heart development upon *Prdm16* deletion, or could be a primary response due to the *Prdm16* deletion as shown by Cibi *et al.* (Cibi DM et al, 2020). However, we do not believe this hypertrophy is part of the phenotypic switch of ventricular towards more atrial or conduction, since in mice ventricular cardiomyocytes are larger in size than atrial (Bogeholz N et al, 2018) and conduction cardiomyocytes (Mandla R et al, 2021), which is in accordance with the function of the ventricles to provide enough force to ensure sufficient perfusion of blood. We have expanded the methods section (lines 821-826) with the new experiments and updated **Table 2**.

2. Figure 1C uses *Nppa* / *Nppb* expression as a proxy for cardiac stress, however these are also markers for unstressed atria. Since later in the paper you dive into this atrial-shift further, why would these markers be a measure of cardiac stress?

Response: While *Nppa* and *Nppb* are frequently used as cardiac stress markers (including in previous papers reporting on *PRDM16* deficiency in cardiomyocytes) (Cibi DM et al, 2020, Man JCK et al, 2021, Sun B et al, 2023), both markers are highly expressed in atrial and ventricular embryonic cardiomyocytes. However, only *Nppa* expression completely diminishes from adult ventricular cardiomyocytes, while *Nppb* continues to be expressed in both ventricular and atrial cardiomyocytes (Man JCK et al, 2021). We concur that in the case of *Nppa*, being also a marker of postnatal atrial cardiomyocytes, it may be confusing to use it as a cardiac stress marker in the context of our phenotype where we see a switch towards an atrial gene signature. In favor of our interpretation that the increased *Nppa* expression levels in *Prdm16* cKO hearts are not solely the cause of a shift to atrial fate, we show here for the Reviewer's interest that the increase caused by *Prdm16* deletion in cardiomyocytes is several-fold higher than the mere difference in expression between ventricular and atrial cardiomyocytes (**Figure R5**). We hence

believe that cardiac stress is a major – if not the main – contributor to the elevated levels of *Nppa* (and *Nppb*).

Figure R5. *Nppa* expression levels in WT or cKO ventricular samples and WT atrial samples.

- Figure 1D shows an increase in fibrosis in the cKO. Why would the hearts develop fibrosis without any injury or excessive stress?

Response: We believe fibrosis is directly related to the increased population of fibroblasts that acquire a more myofibroblast-like gene signature, as we point out in our answer to Question 2 of Reviewer 2. Although the hearts of *Prdm16* mutants likely did not develop ischemic/hypoxic stress causing massive cardiomyocyte death and scar formation (which is also in accordance with the troponin levels that were not notably increased; data not shown), there was a loss of cardiomyocytes (as shown by the decreased proportion from 48% to 33% in Figure 2 of the original manuscript, and now also confirmed *in situ* by IHC analysis as an answer to your Question 6) after which the lost muscle tissue likely is replaced by myofibroblasts that deposit a collagen-rich extracellular matrix. Cardiomyocyte death in this case is likely the consequence of the increased workload on the cardiomyocytes due to abnormal development and specification of the working myocardium.

- Figure 1E shows immunofluorescence stainings of endomucin. Due to the high intensity detected outside of the endocardium, is this marker the best way to determine the trabeculae/lumen interface? The lines drawn do not clearly correspond to the detected. The endomucin signal seems stronger in the cKO, are these imaged at the same settings? Could the slight increase in endothelial cells explain this difference? Furthermore, the way the compact layer thickness is quantified is unclear.

Response: As mentioned in our answer to Question 6 of Reviewer 2, we acknowledge that ENDOMUCIN is not a specific endocardial marker, even though it has been used by others to delineate the endocardium and hence guide the analysis of trabeculae formation and/or thickness of the compact layer (Feng Q et al, 2010, Liu H et al, 2023, Lu P et al, 2023, Sun B et al, 2023, Wu T et al, 2022). In order to confirm the data based on ENDOMUCIN, we repeated the analysis thereby using a marker that is more specific for endocardium, *i.e.* NPR3. We have replaced the ENDOMUCIN pictures with NPR3 pictures in revised Figure 1E and included a new quantification based on NPR3 stainings. As additional validation, we have now moved the

ENDOMUCIN-based analysis to the supplement (Figure S3) of the revised manuscript. As the Reviewer can appreciate, the outcome of the NPR3-based analysis is very similar to that based on ENDOMUCIN. For clarity, we have removed the purple lines from the original ENDOMUCIN figure that were intended to show the endocardial lining. Instead, we now indicate both on the ENDOMUCIN and NPR3-based images the border between the compact and trabecular zone. We concur with the Reviewer that the increase in endocardial ENDOMUCIN signal is due to the increased number of endocardial endothelial cells we reported in **Figure S3**. This increased signal was also apparent for NPR3, as we show in revised **Figure 1E**. We apologize for not being sufficiently clear on how the analysis of compact layer thickness was performed. We have expanded the methods section with information on the NPR3 staining (lines 836 and 840-847), provided more analysis details in the methods section of revised manuscript (lines 850-851) and adapted the figure legend (lines 1165, 1168).

5. Figure 1A shows the UMAP of the integrated datasets, however perhaps a UMAP with the separate replicates could be added to confirm the clustering is not due to inter-replicate variability.

Response: We thank the Reviewer for this valuable remark. We apologize for the confusion. The term 'integrated' refers in this case to the integration of the ATACseq and RNAseq datasets, which originate from the same nuclei but required a different QC approach upon integrating. Hence integration does refer to the integration of biological replicates for each dataset. Indeed, due to the limited yield of cells from the left ventricle of a P7 heart, sequencing was performed in such a way that all 4 *WT* replicates were pooled and sequenced in one lane, and all 4 *cKO* replicates were pooled and sequenced in a second lane. There was no extra cell-hashing or barcoding added to the replicates to pull these apart in our data analysis. However, as an alternative way to support the notion that the cluster shift within the cardiomyocyte population was not simply related to inter-replicate variability, we have added an extra diagram to revised **Figure 2A**, showing a UMAP of our cellular landscape split in *WT versus cKO* and adapted the figure legend (lines 1179-1180). We hope it is clear to the Reviewer that all other cell types (fibroblasts, endothelial cells, mural cells and immune cells) cluster together independent of genotype, while only for the cardiomyocyte *cKO* and *WT* cell clusters do not entirely overlap. Moreover, also the proliferating cardiomyocytes cluster together independent of genotypes (**Figure 2A**).

6. Figure 1C shows that there is a shift in the fractions of different cell types present in the *cKO* when compared to the fractions in the *WT*. However, it is best to be careful with quantifications like these because the isolation has a strong impact on which cell types are preferentially isolated. The same protocol on the *WT* and the *cKO* could lead to different fractions purely due to the isolations. Could an experiment be added that quantifies cell types using immunofluorescent stainings?

Response: We agree with the Reviewer that the isolation procedure can have a significant impact on the cellular landscape determined based on our sequencing data as there may be a cell-type specific vulnerability to the dissociation protocol used. This is why we took extra precautions in order to limit bias in cellular composition due to the isolation procedure. First, samples were processed for sequencing in the same way and on the same day. Thus, if the protocol would influence specific cell types in sample one, it would have also influenced that same cell type in sample 2 since the same protocol was used. From the start of the isolation, samples were manipulated equally (QC on the same day, library preparation on the same day, sequencing on

the same day and on the same chip) to minimize different impacts on the samples. Second, because we used nuclei instead of cells, the isolation-related bias is a much lesser concern, since nuclei are more resistant to harsh digestion conditions (Wu H et al, 2019). Nevertheless, despite these precautions, we concur that using a method that allows to quantify cell numbers *in situ* in the tissue (*i.e.* by immunostaining for cell-specific markers) would offer a way to perform quantification without any isolation-related bias. To show that the proportions obtained by omics data and those retrieved from immunostaining are indeed in accordance, we already had one example in our original manuscript where we show a 4.6-fold (**Figure S4D**) and 5.4-fold (revised **Figure 4D**) increase in Purkinje fibers measured by omics and immunostaining, respectively. The *in situ* quantifications based on immunofluorescence confirmed our omics-based analysis for cardiomyocytes as well. These additional data related to *in situ* assessment of cell population shifts have been added to the revised manuscript (lines 235-236) and in revised **Figure 2D**. The figure legend was updated accordingly (lines 1196-1199).

7. The study provides data showing that PRDM16 loss in cardiomyocytes triggers changes in gene expression and chromatin accessibility. The data show that the 'proliferative' cardiomyocytes overlap between the WT and the cKO, but the cluster of 'mature' cardiomyocytes is greatly diminished in the cKO. These data support the role of PRDM16 in regulating key pathways involved in cardiac function and cellular identity. The m+pCM nomenclature was taken from the clusters found in figure 2, however since this zoomed-in clustering still results in a mature and a proliferative cluster, this nomenclature is unclear

Response: We apologize for the confusion and have explained the pooling of our cardiomyocyte clusters better in the revised version of the manuscript. To avoid further confusion, where appropriate, the term 'm+pCM' is systematically replaced in all main (**Figure 3 and 4**) and supplementary Figures (**Figure S5**) by 'CM'.

8. All clusters of cardiomyocytes of the WT and the cKO are pooled to perform DEG analysis. This would be expected to dilute differential signals by adding in the cells with a similar identity (cluster 2 and part of cluster 1). Wouldn't it be interesting to compare cluster 0 and cluster 1 (perhaps with the WT purkinje cells excluded). Further, you mention hyperplasia of the ventricular conduction system, could you perform a direct comparison between the cluster 1 cells from the cKO and the very small portion of cluster 1 WT cells that are supposed to represent the VCS cells in a healthy mouse. This way you could perhaps identify in what way these cells are and are not similar to the Purkinje fibres.

Response: We thank the Reviewer for the suggestion regarding the DEG comparisons. For the Reviewer's interest, we have generated the suggested DEG list resulting from a comparison between cluster 0 WT cells with cluster 1 cKO cells (1,410 DEGs) and overlapped it with the DEG list resulting from our current comparative approach based on the pooled clusters per genotype (1,665 DEGs). We hope the Reviewer appreciates that given the substantial overlap (85%, 1,210 common DEGs), the obtained DEG list is very similar to the one obtained when pooling all sub-clusters together for each genotype, meaning that the contributions of cKO cells present in cluster 0 and WT cells present in cluster 1 or cells present in cluster 2 had a very minor impact on the DEG analysis. We made a note in the revised results text (lines 281, 282). The reason we performed DEG analysis of total cKO versus total WT cardiomyocyte clusters was to look at a more global change at low resolution because of the drastic identity switch. We

believe that the more nuclei we could include in the analysis, the more stringent our DEG analysis would be.

We agree and thank the Reviewer for the second suggestion of comparing the *cKO* sub-cluster 1 cells with the *WT* sub-cluster 1 cells, with the intention to estimate how close the sub-cluster 1 *cKO* cells would come to *bona fide* VCS cells. However, we need to mention here again our wrongful labeling of sub-cluster 1 with overrepresented *cKO* nuclei as a 'VCS' cluster. Indeed, even though this sub-cluster has a gene signature that is highly enriched for genes known to be expressed in the VCS, the same sub-cluster also shows enrichment for genes known to be associated with atrial cardiomyocytes. Furthermore, many of the genes within the atrial signature overlap with genes in the VCS signature, as we point out in our answer to your Question 10. Given the mixed nature of the expression shifts in this sub-cluster, we should not merely label it as a VCS cluster, but rather as 'mixed'. We hence systematically revised the manuscript to avoid such a misleading statement (lines 244-247, 269, 277, 278; Figure S5A now shows a more appropriate set of marker genes comprising both atrial and conduction markers).

Finally, we do not expect that comparing *WT* cluster 1 cells with *cKO* cluster 1 cells would offer the possibility of knowing whether the cluster 1 *cKO* cells are similar to Purkinje fibers, since Purkinje fibers are only a small subset of the VCS, as we point out in our answer to Question 4 of Reviewer 2. Within cluster 1, by further clustering at higher resolution, we did identify a pure Purkinje fiber subpopulation (cluster 7 in Figure S3 of the original manuscript). Due to the expression of *Cntn2*, we could identify this population as a pure Purkinje fiber cluster, which truly expanded due to PRDM16 loss, as shown by our CONTACTIN-2 staining data. Hence, we reworded the VCS hyperplasia we refer to in the manuscript as 'distal VCS hyperplasia', which corresponds to the Purkinje fibers (lines 53, 131, 382, 513, 549, 852; Figure 7 and graphical abstract).

9. Rather than using upregulated and downregulated for the volcano plots, use higher and lower expressed. Regulation implies an active process which, while it's possible that this is the case here, is not shown using two static captures of separate transcriptomes. Clearly communicate that each volcano plot is the same plot, and do not use different a x-axis for one of them. The number of DEGs found to be lower expressed in the *cKO* is lower, therefore it is difficult to assess whether the fraction of genes for the gene-sets tested is actually that different. Could you add a functional annotation of KEGG pathways or a different measure, and perform the statistical enrichment.

Response: We thank the Reviewer for the remark regarding **terminology**. We agree that 'up- or downregulated' implies a time dimension in the comparison and that in the absence thereof 'higher expressed' or 'lower expressed' is more appropriate. We have adapted this systematically in the revised manuscript (lines 285, 288, 290, 292-294, 323, 325, 348, 411, 1207, 1208, 1213, 1215, 1233, 1234, 1237, 1238).

We thank the Reviewer for the comment of using the same x-axis for each **Volcano plot**, as they indeed originate from the same comparison. We have adapted Figure 3C accordingly. We concur with the Reviewer's comment that given the overall different number of DEGs between higher or lower expressed, it is more appropriate to use proportions instead of absolute numbers of genes in the Volcano plots. Therefore, in addition to the absolute number of genes we now also mention the proportions in Figure 3C and we have adapted the results text (lines 316-318) and figure legend (line 1210) accordingly.

As requested, a **functional annotation** and statistical enrichment was added to supplementary Table S6 and integrated in the revised results section of the manuscript (lines 331-336). To perform this functional annotation, we used ToppGene, also used by Wu *et al.* (Wu T *et al.*, 2022), to be able to also look for similarities between both *Prdm16* cKO models for the Reviewer's and readers' interest. Terms related to our phenotype are highlighted in red. The methods text was also updated (lines 986-987).

10. Figure 4A shows the expression pattern of certain genes classified as atrial to be expressed in the small population of Purkinje cells of the WT (e.g. *Myl4*, *Fgf12*). Could more convincing data be presented to imply that this is a shift to both atrial and VCS, rather than just VCS. Seemingly there is overlap between atrial and VCS specific properties. Furthermore, could also be elaborated on the ways these cells are not similar to healthy WT Purkinje cells and atrial cells (although that would understandably be more complicated due to the lack of atrial cells in the dataset)?

Response: We acknowledge the Reviewer's comment that there is indeed overlap between the atrial and conduction reference signatures (which we retrieved from two previously published papers (Cao Y *et al.*, 2023, Shekhar A *et al.*, 2018)). This means that the shifts we show in the top panels of **Figure 4C,D** of the revised manuscript encompass both markers unique or common to either atrial or conduction fate. In order to provide additional evidence that there is indeed also a ventricular-to-atrial shift – and not merely a shift towards conduction fate – we repeated the analysis shown in **Figure 4C,D**, but now we only used genes that were unique for atrial or conduction by filtering out the genes common to both atrial and conduction gene signatures. As you can appreciate, we see very similar shifts as in **Figure 4C,D**. For reasons of clarity, we have reworded the paragraph reporting the double shift and commented on this issue of overlapping signatures in the revised text (lines 346-361) and added this complementary analysis to revised Figure S6A,B. Furthermore, the full list of unique atrial or conduction genes shown in **Figure S6A,B** have been added to revised **Table S7**. Since there is a double fate shift, the majority of the cells likely acquire a mixed gene signature such that they are not identical to wild-type ventricular conduction cardiomyocytes or wild-type atrial cardiomyocytes.

11. Figure 4H shows two images of a *Cntn2* staining, which is nicely restricted to the Purkinje. However, do these sections represent a similar depth cut through the heart. It could be contractile state, but it looks like the cavity size is small due to being a fairly shallow cut in the WT. Furthermore, the quantification looks much too high, the cKO image does not show 10% of the total ventricular area to be *Cntn2* positive. It does look like an increase, but roughly 10% sounds very unlikely. Did the measurement take the folds with strong background into account?

Response: We apologize for this oversight and have replaced the panels with more representative images of the same anatomical location. We do see ventricular dilation in the *Prdm16* mutant hearts, hence their ventricular cavities are larger. To avoid bias in quantification due to not being in the same anatomical location, we performed the quantification for each mouse heart on several serial sections and calculated the average across these sections to obtain the final area%. While analyzing the images we manually excluded the artifacts caused by section folding. As we are aware that performing morphometry on sections based on immunofluorescence images is subjective and observer-dependent, the analysis was repeated by another independent researcher who obtained the same and significant fold-difference between genotypes, but with somewhat lower absolute numbers which may better correspond

to the representative images shown. We have now added the new analysis to the revised results text (line 377) and revised Figure 4F.

12. Finally, in the discussion (line 429) the following is written "in CMs we did not see elaborate transcriptomic signs of a switch towards trabecular fate upon PRDM16 loss". Could a figure be added that shows this observation?

Response: We concur with the Reviewer that we should add data to our manuscript to support our claim related to the trabecular fate shift. We acknowledge that perhaps our statement 'not seeing elaborate transcriptomic signs of a switch towards trabecular fate' was somewhat misleading since only some (*e.g.*, *Thbs4*) – but definitely not all – trabecular markers were indeed higher expressed in *Prdm16* mutant hearts and some (*e.g.*, *Hey2*) – but definitely not all – compact markers were indeed lower expressed. This higher expression/lower expression, respectively, was expected based on the paper by Wu *et al.* (Wu T *et al.*, 2022) but it was significantly less prominent in our case where we looked at the postnatal heart. Overall, for the Reviewer's interest, only 28% of the top 50 trabecular or compact marker genes taken from Li *et al.* (Li G *et al.*, 2016) were altered in the expected direction. Therefore, we have reworded the paragraph discussing the impact of the time window of analysis on the cardiomyocyte fate shift and we particularly rephrased our original statement on the trabecular shift in the revised manuscript (lines 568-600) to more accurately reflect this less prominent shift and we now support the latter by referring to a newly added supplementary Figure S10, where we show the expression of a panel of trabecular and compact marker genes, the majority of which being unaltered.

13. Line 314 states that "PRDM16 puts a double brake on their activity by lowering both their expression and accessibility to DNA" however, doesn't lowered accessibility necessarily decrease the expression anyway, how can these two mechanisms be separated?

Response: We apologize for the confusing statement. We agree that when the accessibility of the regulatory regions in the DNA of a target gene is lowered for transcription factors, that then this would lead to lower expression of the target gene. However, the expression we refer to in our statement is not that of the targets of the transcription factors but rather that of the transcription factors themselves. We have rephrased the statement to clarify this (lines 421-422).

14. For the heatmaps in figure 5D and E it is unclear what exactly these represent. It would benefit from a more comprehensive explanation as to what the axes represent and which variable is presented by the colour-scale.

Response: We apologize for being insufficiently clear. In the revised manuscript we have adapted Figure 5 by adding clear labeling of the axes and we have reworded the results text (lines 423-441) and the figure legend (lines 1252, 1256-1257, 1261-1292) to make it more comprehensive and provide a better explanation as to what they exactly represent.

15. Figure 6A shows a decision tree for identifying mechanisms of action for PRDM16. Why was the ChIP-seq data not also checked for signal in the identified distal enhancer elements?

Response: We concur with the Reviewer's question. Even though we intended to also consider enhancer regions to look for PRDM16 binding sites, this was unfortunately not possible. We used publicly available ChIPSeq data generated by Wu *et al.* (Wu T et al, 2022), which only included promotor regions. We have contacted the authors to acquire the full list of differential peaks to also identify and look at enhancers. They have provided us the BigWig files to look at the racks in Integrated Genome Viewer, however, it is not common to use these BigWig files for peak quantifications since these files lack counts. Hence, the raw information is lost by generating BigWig files. Furthermore, since they did not perform HiC in addition to the PRDM16 ChIPseq, it was not possible to annotate enhancer regions (which is the reason why also they did not include enhancer analysis in their paper). Therefore, in our decision tree it is only possible to include PRDM16-ChIPseq promotor regions of this dataset.

REFERENCES in REBUTTAL

- Bogeholz N, Pauls P, Dechering DG, Frommeyer G, Goldhaber JI, Pott C, Eckardt L, Muller FU, Schulte JS (2018) Distinct occurrence of proarrhythmic afterdepolarizations in atrial versus ventricular cardiomyocytes: Implications for translational research on atrial arrhythmia. *Front Pharmacol* 9: 933. doi:10.3389/fphar.2018.00933
- Cao Y, Zhang X, Akerberg BN, Yuan H, Sakamoto T, Xiao F, VanDusen NJ, Zhou P, Sweat ME, Wang Y, et al (2023) In vivo dissection of chamber-selective enhancers reveals estrogen-related receptor as a regulator of ventricular cardiomyocyte identity. *Circulation* 147: 881-896. doi:10.1161/CIRCULATIONAHA.122.061955
- Cibi DM, Bi-Lin KW, Shekeran SG, Sandireddy R, Tee N, Singh A, Wu Y, Srinivasan DK, Kovalik JP, Ghosh S, et al (2020) Prdm16 deficiency leads to age-dependent cardiac hypertrophy, adverse remodeling, mitochondrial dysfunction, and heart failure. *Cell Rep* 33: 108288. doi:10.1016/j.celrep.2020.108288
- Craps S, Van Wauwe J, De Moudt S, De Munck D, Leloup AJA, Boeckx B, Vervliet T, Dheedene W, Criem N, Geeroms C, et al (2021) Prdm16 supports arterial flow recovery by maintaining endothelial function. *Circ Res* 129: 63-77. doi:10.1161/CIRCRESAHA.120.318501
- Fei LR, Huang WJ, Wang Y, Lei L, Li ZH, Zheng YW, Wang Z, Yang MQ, Liu CC, Xu HT (2019) Prdm16 functions as a suppressor of lung adenocarcinoma metastasis. *J Exp Clin Cancer Res* 38: 35. doi:10.1186/s13046-019-1042-1
- Feng Q, Di R, Tao F, Chang Z, Lu S, Fan W, Shan C, Li X, Yang Z (2010) Pdk1 regulates vascular remodeling and promotes epithelial-mesenchymal transition in cardiac development. *Mol Cell Biol* 30: 3711-3721. doi:10.1128/MCB.00420-10
- Kramer RJ, Fatahian AN, Chan A, Mortenson J, Osher J, Sun B, Parker LE, Rosamilia MB, Potter KB, Moore K, et al (2023) Prdm16 deletion is associated with sex-dependent cardiomyopathy and cardiac mortality: A translational, multi-institutional cohort study. *Circ Genom Precis Med* 16: 390-400. doi:10.1161/CIRCGEN.122.003912
- Kuhnisch J, Theisen S, Dartsch J, Fritsche-Guenther R, Kirchner M, Obermayer B, Bauer A, Kahlert AK, Rothe M, Beule D, et al (2024) Prdm16 mutation determines sex-specific cardiac metabolism and identifies two novel cardiac metabolic regulators. *Cardiovasc Res* 119: 2902-2916. doi:10.1093/cvr/cvad154
- Li G, Xu A, Sim S, Priest JR, Tian X, Khan T, Quertermous T, Zhou B, Tsao PS, Quake SR, et al (2016) Transcriptomic profiling maps anatomically patterned subpopulations among single embryonic cardiac cells. *Dev Cell* 39: 491-507. doi:10.1016/j.devcel.2016.10.014
- Liu H, Duan R, He X, Qi J, Xing T, Wu Y, Zhou L, Wang L, Shao Y, Zhang F, et al (2023) Endothelial deletion of ptbp1 disrupts ventricular chamber development. *Nat Commun* 14: 1796. doi:10.1038/s41467-023-37409-9
- Lu P, Wu B, Wang Y, Russell M, Liu Y, Bernard DJ, Zheng D, Zhou B (2023) Prerequisite endocardial-mesenchymal transition for murine cardiac trabecular angiogenesis. *Dev Cell* 58: 791-805 e794. doi:10.1016/j.devcel.2023.03.009
- Man JCK, van Duijvenboden K, Krijger PHL, Hooijkaas IB, van der Made I, de Gier-de Vries C, Wakker V, Creemers EE, de Laat W, Boukens BJ, et al (2021) Genetic dissection of a super enhancer controlling the nppa-nppb cluster in the heart. *Circ Res* 128: 115-129. doi:10.1161/CIRCRESAHA.120.317045

- Mandla R, Jung C, Vedantham V (2021) Transcriptional and epigenetic landscape of cardiac pacemaker cells: Insights into cellular specialization in the sinoatrial node. *Front Physiol* 12: 712666. doi:10.3389/fphys.2021.712666
- Nam JM, Lim JE, Ha TW, Oh B, Kang JO (2020) Cardiac-specific inactivation of prdm16 effects cardiac conduction abnormalities and cardiomyopathy-associated phenotypes. *Am J Physiol Heart Circ Physiol* 318: H764-H777. doi:10.1152/ajpheart.00647.2019
- Shekhar A, Lin X, Lin B, Liu FY, Zhang J, Khodadadi-Jamayran A, Tsirigos A, Bu L, Fishman GI, Park DS (2018) Etv1 activates a rapid conduction transcriptional program in rodent and human cardiomyocytes. *Sci Rep* 8: 9944. doi:10.1038/s41598-018-28239-7
- Shimada IS, Acar M, Burgess RJ, Zhao Z, Morrison SJ (2017) Prdm16 is required for the maintenance of neural stem cells in the postnatal forebrain and their differentiation into ependymal cells. *Genes Dev* 31: 1134-1146. doi:10.1101/gad.291773.116
- Strassman A, Schnutgen F, Dai Q, Jones JC, Gomez AC, Pitstick L, Holton NE, Moskal R, Leslie ER, von Melchner H, et al (2017) Generation of a multipurpose prdm16 mouse allele by targeted gene trapping. *Dis Model Mech* 10: 909-922. doi:10.1242/dmm.029561
- Sun B, Rouzbehani OMT, Kramer RJ, Ghosh R, Perelli RM, Atkins S, Fatahian AN, Davis K, Szulik MW, Goodman MA, et al (2023) Nonsense variant prdm16-q187x causes impaired myocardial development and tgf-beta signaling resulting in noncompaction cardiomyopathy in humans and mice. *Circ Heart Fail* 16: e010351. doi:10.1161/CIRCHEARTFAILURE.122.010351
- Wu H, Kirita Y, Donnelly EL, Humphreys BD (2019) Advantages of single-nucleus over single-cell mass sequencing of adult kidney: Rare cell types and novel cell states revealed in fibrosis. *J Am Soc Nephrol* 30: 23-32. doi:10.1681/ASN.2018090912
- Wu T, Liang Z, Zhang Z, Liu C, Zhang L, Gu Y, Peterson KL, Evans SM, Fu XD, Chen J (2022) Prdm16 is a compact myocardium-enriched transcription factor required to maintain compact myocardial cardiomyocyte identity in left ventricle. *Circulation* 145: 586-602. doi:10.1161/CIRCULATIONAHA.121.056666

September 4, 2024

RE: Life Science Alliance Manuscript #LSA-2024-02719-TR

Prof. Aernout Luttun
KU Leuven
Department of Cardiovascular Sciences Center for Molecular and Vascular Biology, Endothelial Cell Biology Unit
KU Leuven, Campus Gasthuisberg Onderwijs & Navorsing 1, Herestraat 49 box 911
Leuven B-3000
Belgium

Dear Dr. Luttun,

Thank you for submitting your revised manuscript entitled "PRDM16 determines specification of ventricular cardiomyocytes by suppressing alternative cell fates". We would be happy to publish your paper in Life Science Alliance pending final revisions necessary to meet our formatting guidelines.

- please be sure that the authorship listing and order is correct
- please upload your supplemental figures as single files
- please add your supplemental figure legends to the main manuscript text
- please make sure that your supplemental tables are uploaded as editable doc or excel files
- thank you for providing a graphical abstract; for graphical abstracts, we do not need a figure legend, so please remove the figure legend 7
- please add a panel E to the figure 5 legend

LSA now encourages authors to provide a 30-60 second video where the study is briefly explained. We will use these videos on social media to promote the published paper and the presenting author (for examples, see <https://docs.google.com/document/d/1-UWCfbE4pGcDdcgzcmiuJI2XMBJnxKYeqRvLLrLS08s/edit?usp=sharing>). Corresponding or first-authors are welcome to submit the video. Please submit only one video per manuscript. The video can be emailed to contact@life-science-alliance.org

A. FINAL FILES:

B. MANUSCRIPT ORGANIZATION AND FORMATTING:

Sincerely,

Please find enclosed our format revisions for manuscript (N° LSA-2024-02719-TR) entitled 'PRDM16 determines specification of ventricular cardiomyocytes by suppressing alternative cell fates.' by Jore Van Wauwe, Alexia Mahy, Sander Craps, Samaneh Ekhteraei-Tousi, Pieter Vrancaert, Hannelore Kemps, Wouter Dheedene, Rosa Doñate Puertas, Sander Trenson, H Llewelyn Roderick, Manu Beerens, and myself for publication in *Life Science Alliance* as an Article. We would like to thank you for the offer to publish our work. Below, we provide an answer to your questions/requests related to format requirements:

-please be sure that the authorship listing and order is correct:

Answer: All authorship data is now correct, one author (M. Beerens) has updated his author account as there was a typing error in his surname.

-please upload your supplemental figures as single files:

Answer: Supplemental figures have now been provided as single pdf files.

-please add your supplemental figure legends to the main manuscript text:

Answer: Legends to supplementary figures have now been added after the main figure legends in the main manuscript.

-please make sure that your supplemental tables are uploaded as editable doc or excel files:

Answer: All supplementary Tables have been uploaded as doc file (Tables S1 and S2) or excel file (Tables S3-S9). The corresponding legend texts for the tables in excel format have been added as an additional tab in each excel file.

-thank you for providing a graphical abstract; for graphical abstracts, we do not need a figure legend, so please remove the figure legend 7:

Answer: We actually intended to use the graphical abstract as Figure 7 as well, but perhaps that makes Figure 7 a bit redundant. Therefore, we decided to only use it as graphical abstract and remove the reference to Figure 7 in the discussion and also remove the figure legend.

-please add a panel E to the figure 5 legend:

Answer: We apologize for this oversight, this has been amended.

We hope our manuscript in its reformatted form is now acceptable for publication in *Life Science Alliance*,

September 9, 2024

RE: Life Science Alliance Manuscript #LSA-2024-02719-TRR

Prof. Aernout Lutun
KU Leuven
Department of Cardiovascular Sciences Center for Molecular and Vascular Biology, Endothelial Cell Biology Unit
KU Leuven, Campus Gasthuisberg Onderwijs & Navorsing 1, Herestraat 49 box 911
Leuven B-3000
Belgium

Dear Dr. Lutun,

Thank you for submitting your Research Article entitled "PRDM16 determines specification of ventricular cardiomyocytes by suppressing alternative cell fates". It is a pleasure to let you know that your manuscript is now accepted for publication in Life Science Alliance. Congratulations on this interesting work.

DISTRIBUTION OF MATERIALS:

Again, congratulations on a very nice paper. I hope you found the review process to be constructive and are pleased with how the manuscript was handled editorially. We look forward to future exciting submissions from your lab.

Sincerely,
